# Understanding and Mitigating Bottlenecks of State Space Models through the Lens of Recency and Over-smoothing

**Peihao Wang[1], Ruisi Cai[1], Yuehao Wang[1], Jiajun Zhu[1,2], Pragya Srivastava[3], Zhangyang Wang[1], Pan Li[4]**
[1]University of Texas at Austin, [2]Zhejiang University, [3]Google DeepMind, [4]Georgia Tech
`{peihaowang,ruisi.cai,yuehao,atlaswang}@utexas.edu, junnian@zju.edu.cn,`
`pragya8srivastava@gmail.com, panli@gatech.edu`

## Abstract

Structured State Space Models (SSMs) have emerged as alternatives to transformers. While SSMs are often regarded as effective in capturing long-sequence dependencies, we rigorously demonstrate that they are inherently limited by strong recency bias. Our empirical studies also reveal that this bias impairs the models' ability to recall distant information and introduces robustness issues. Our scaling experiments then discovered that deeper structures in SSMs can facilitate the learning of long contexts. However, subsequent theoretical analysis reveals that as SSMs increase in depth, they exhibit another inevitable tendency toward over-smoothing, e.g., token representations becoming increasingly indistinguishable. This *fundamental dilemma* between recency and over-smoothing hinders the scalability of existing SSMs. Inspired by our theoretical findings, we propose to *polarize* two channels of the state transition matrices in SSMs, setting them to zero and one, respectively, simultaneously addressing recency bias and over-smoothing. Experiments demonstrate that our polarization technique consistently enhances the associative recall accuracy of long-range tokens and unlocks SSMs to benefit further from deeper architectures. All source codes are released at `https://github.com/VITA-Group/SSM-Bottleneck`.

## 1 Introduction

Sequence processing architectures have evolved from RNNs (Hochreiter & Schmidhuber, 1997; Sutskever et al., 2014; Cho et al., 2014; Cho, 2014), to transformers (Vaswani et al., 2017; Devlin et al., 2019; Radford et al., 2018; 2019; Brown et al., 2020), and more recently to State Space Models (SSMs) (Gu et al., 2021a; Gu & Dao, 2023). In particular, SSMs (Gu et al., 2021a; Gu & Dao, 2023; Dao & Gu, 2024) have emerged as a compelling alternative to transformers, enabling more efficient handling of long sequences. SSMs operate in two modes: convolution and recurrence (Gu et al., 2021b). During convolutional mode, SSMs assume visibility of the entire sequence and utilize hardware-optimized convolutions to propagate information across all tokens in parallel. This approach avoids the calculation of pairwise correlations inherent in attention mechanisms, thereby accelerating training. Mamba (Gu & Dao, 2023) enabled convolution with a parallel scanning algorithm, which yielded more expressive sequence-level mixing without sacrificing efficiency. During recurrent mode, SSMs process one token at a time while maintaining a compact recurrent hidden state encoding the sequence history. The outputs are sequentially decoded from this hidden state, avoiding storing all past key-value pairs (Dai et al., 2019) and thus reducing inference memory usage.

Furthermore, SSMs have been meticulously tailored to effectively capture long-range dependencies and filter contextual information. These models are grounded in HiPPO theory (Gu et al., 2020), which demonstrates that a first-order Ordinary Differential Equation (ODE) can encapsulate long-term memory through a designated state matrix known as the HiPPO matrix. Subsequent research (Gu et al., 2021b;a; Gupta et al., 2022; Gu et al., 2022a) has simplified this state matrix to a diagonal form, significantly improving computational efficiency while retaining the ability to model long-range dependencies. Mamba (Gu & Dao, 2023) introduced a selection mechanism that selectively aggregates pertinent information from the context into the state. Concurrently, linear attention models

have been derived from streamlined attention mechanisms (Katharopoulos et al., 2020; Sun et al., 2023; Peng et al., 2023; Yang et al., 2023). Collectively, these advances can be interpreted through a unified lens as more structured SSMs (Dao & Gu, 2024).

Despite their initial empirical successes, recent findings indicate that SSMs may not match transformers in their ability to recall information from long contexts (Arora et al., 2023; Poli et al., 2024) or in handling more complex retrieval patterns (Park et al., 2024). Furthermore, it has been observed that Mamba continues to underperform compared to transformers on larger scales (Waleffe et al., 2024). These shortcomings have not yet been systematically elucidated.

In this paper, we identify two fundamental limitations of SSMs in their ability to model complex long-range dependencies. First, we argue that the long-term memory capabilities of modern SSMs may be misinterpreted. Our analysis reveals that an SSM layer exhibits a strong recency bias, limiting tokens to primarily interact with the nearby context. This bias is intrinsic to SSMs and many linear attention models, regardless of the content-informing techniques employed, such as the selection mechanism introduced by Mamba (Gu & Dao, 2023). We further posit that the loss of long-range capabilities may stem from the oversimplification of HiPPO-induced SSMs, which trades efficiency off performance. To substantiate this claim, we perform a long-range retrieval task on an industrial-scale language model (Jiang et al., 2023) based on Mamba. Our test results indicate that Mamba catastrophically forgets distant content once the context length exceeds its memory capacity. Furthermore, we raise a novel robustness concern regarding SSMs with recency bias: our empirical outcomes show that Mamba is more susceptible to perturbations on local tokens, making it vulnerable to adversarial attack, as these local tokens can be easily manipulated to serve as backdoors.

We then conduct a series of scaling experiments with varying context lengths during the pre-training of SSMs. Our results indicate that increasing the model depth is crucial for enhancing its ability to utilize long contexts by expanding the receptive field. However, we observe that depth scaling encounters another bottleneck as performance begins to saturate as depth continues to increase. We analyze the feature dynamics across the SSM layers and theoretically revealed that SSMs inherently function as smoothing operators, leading to over-smoothing in deep architectures (NT & Maehara, 2021; Oono & Suzuki, 2019; Cai & Wang, 2020). As a result, token representations become increasingly uniform and indistinguishable with each additional layer.

Recognizing such fundamental dilemma between recency and over-smoothing, we introduce a unified approach called the *polarization*, which simultaneously addresses recency bias and over-smoothing. Specifically, we reserve two dedicated channels in the state transition matrices of SSMs and reset them to zero and one, respectively. The all-one channel helps preserve historical information and prevents catastrophic forgetting caused by locality, while the zero channel inhibits excessive fusion of information from past tokens, effectively slowing the smoothing rate. Our experiments on associative recall (Arora et al., 2023) demonstrate that the polarization technique significantly improves recall accuracy by mitigating locality artifacts and can gain more performance when combined with deeper architectures by effectively alleviating over-smoothing.

## 2 PRELIMINARIES

In SSMs (and alike), we represent the a discrete-time sequence of $T$ tokens as $\boldsymbol{x} = [\boldsymbol{x}_1 \ \cdots \ \boldsymbol{x}_T]^\top \in \mathbb{R}^T$. For vector-valued input sequences, SSMs process each channel independently. To simplify notations, we focus on scalar-valued sequences without loss of generality. The impact of multi-channel inputs will be addressed in the relevant context. SSMs learn to represent and forecast the next token by integrating past information. Formally, SSMs can be viewed as a sequence-to-sequence transformation from inputs $\boldsymbol{x} \in \mathbb{R}^T$ to outputs $\boldsymbol{y} \in \mathbb{R}^T$ through a *memory state* $\boldsymbol{h}_t \in \mathbb{R}^N$, which is iteratively updated with a linear recurrence. A general form can be written as:

$$\boldsymbol{h}_t = \boldsymbol{A}_t \boldsymbol{h}_{t-1} + \Delta_t \boldsymbol{b}_t(\boldsymbol{x}_t), \quad \boldsymbol{y}_t = c_t(\boldsymbol{h}_t), \quad \boldsymbol{h}_0 = \boldsymbol{0}, \quad \forall t \in [T], \tag{1}$$

where $t \in [T]$ denotes the time step. Intuitively, $\boldsymbol{A}_t \in \mathbb{R}^{N \times N}$ extracts information from the previous state $\boldsymbol{h}_{t-1}$ [1], $\boldsymbol{b}_t : \mathbb{R} \to \mathbb{R}^N$ projects every input token to the hidden space, $\Delta_t \in \mathbb{R}$ controls how much information of the new token will be fused into the hidden memory, and $c_t : \mathbb{R}^N \to \mathbb{R}$ decodes

---

[1]In most scenarios discussed in this paper, we assume real parameterization by default, as it is the standard approach in the cases of our primary interest, such as language modeling (Gu & Dao, 2023)

the hidden state at time $t$ to the final prediction. In all SSMs considered in this paper, it is necessary to assume $(\boldsymbol{A}_t, \boldsymbol{b}_t, c_t, \Delta_t)$ *only depends on the inputs at the time $t$*. SSMs are trained from end to end to optimize for the parameters $\{(\boldsymbol{A}_t, \boldsymbol{b}_t, c_t, \Delta_t)\}_{t \in [T]}$, for which the different SSMs adopt various types of instantiation. Below, we list some representative examples.

**S4, DSS, and S4D.** The seminal works (Gu et al., 2020; 2021b; 2022b) demonstrate that discretizing time-invariant ODE $\boldsymbol{h}'(t) = \boldsymbol{A}\boldsymbol{h}(t) + \boldsymbol{b}x(t)$ with some special realization of matrix $\boldsymbol{A}$ can yield an efficient recurrent network for long-sequence modeling. The follow-up works Gu et al. (2021a) together with Gupta et al. (2022); Gu et al. (2022a) simplifies $\boldsymbol{A}$ to be a diagonal matrix. Applying the zero-order hold rule for discretization, as suggested by Gupta et al. (2022), we can summarize this series of models in the form of Eq. 1:

$$\text{(S4)} \quad \boldsymbol{A}_t = \exp(\Delta \boldsymbol{A}), \quad \boldsymbol{b}_t(\boldsymbol{x}_t) = \boldsymbol{b}\boldsymbol{x}_t, \quad c(\boldsymbol{h}_t) = \boldsymbol{c}^\top \boldsymbol{h}_t, \quad \Delta_t = \Delta,\,^2 \tag{2}$$

where $(\boldsymbol{A}, \boldsymbol{b}, \boldsymbol{c}, \Delta)$ are learnable parameters. In particular, $\boldsymbol{A}$ is restricted to be a diagonal matrix and can be complex valued. However, $\boldsymbol{A}$ must have negative real part (Gu et al., 2022a). $\Delta \in (0, 1]$ is often interpreted as the time interval for discretization. We call this family of SSMs *S4* following the naming convention in Gu & Dao (2023).

**Mamba.** A recent breakthrough Mamba (Gu & Dao, 2023) introduces the *selection* mechanism to extend S4. Instead of learning $(\boldsymbol{A}, \boldsymbol{b}, \boldsymbol{c}, \Delta)$ in Eq. 2 as free parameters, Mamba conditions $(\boldsymbol{A}, \boldsymbol{b}, \boldsymbol{c}, \Delta)$ on the inputs, which enables each iterative step in Eq. 1 to filter useful token information during the recurrence. Specifically, Mamba computes $(\boldsymbol{A}_t, \boldsymbol{b}_t, c_t, \Delta_t)$ as follows:

$$\text{(Mamba)} \quad \boldsymbol{A}_t = \exp(\Delta_t \boldsymbol{A}), \; \boldsymbol{b}_t(\boldsymbol{x}_t) = (\boldsymbol{W}_B \boldsymbol{x}_t)\boldsymbol{x}_t, \; c_t(\boldsymbol{h}) = (\boldsymbol{W}_C \boldsymbol{x}_t)^\top \boldsymbol{h}_t, \; \Delta_t = \sigma(\boldsymbol{W}_\Delta \boldsymbol{x}_t), \tag{3}$$

where $\boldsymbol{W}_\Delta \in \mathbb{R}, \boldsymbol{W}_B \in \mathbb{R}^N, \boldsymbol{W}_C \in \mathbb{R}^N$ are learnable weights in addition to $\boldsymbol{A}$, and $\sigma(\cdot)$ denotes softplus activation. When handling multi-dimensional token embeddings, $\boldsymbol{W}_\Delta, \boldsymbol{W}_B, \boldsymbol{W}_C$ are extended on the input dimension, different $(\boldsymbol{A}_t, \boldsymbol{b}_t, c_t, \Delta_t)$ are assigned to each token channel and Eq. 3 is running channel-wisely. $\Delta_t$ is varying across different channels, while $\boldsymbol{b}_t, \boldsymbol{c}_t$ are shared across token channels. That is, if the token embedding dimension is $D$, then each token channel will be allocated with a distinct $\boldsymbol{W}_\Delta \in \mathbb{R}^D$, and shared $\boldsymbol{W}_B \in \mathbb{R}^{N \times D}$, and $\boldsymbol{W}_C \in \mathbb{R}^{N \times D}$. In language modeling, $\boldsymbol{A}$ has strictly negative real-valued diagonal, which ensures $\boldsymbol{A}_t \in (0, 1)^{N \times N}$. Additionally, Mamba is integrated with the H3 architecture (Fu et al., 2022), wherein the selective SSMs is working with a local convolution and sandwiched by two gated connections.

**Linear Attention.** Concurrent with SSMs, there is another line of work streamlining attention to linear time complexity. With slight abuse of terminology, we name all of them collectively as Linear Attention Models (LAMs). We observe that many of them can be written in the form of Eq. 1 such that in the remainder of this paper, we extend the definition of SSMs to include LAMs without introducing ambiguity, as LAMs and SSMs are dual to each other (Dao & Gu, 2024). We leave a full summary to Appendix B.

## 3 CAN SSM EFFECTIVELY REPRESENT LONG-RANGE DEPENDENCIES?

### 3.1 SSMS ARE LOCALLY BIASED

In this section, we investigate the ability of SSMs to learn long-range dependencies. Recent studies find that SSMs seem more effective than transformers on this task (Gu et al., 2020; Tay et al., 2020; Li et al., 2022; Gu & Dao, 2023). However, in Sec. 3.1 we theoretically show a negative result that an SSM layer is inherent to *local bias* and loses long-term memory exponentially. In Sec. 3.2, we empirically justify our claim by showing SSMs struggle to retrieve from distant context. We also demonstrate that the local bias may lead to robustness issues in Sec. 3.3.

To understand how information is propagated and long-range dependencies are modeled in SSMs, we aim to uncover the relationship between the output at time $t \in [T]$ and the input token at time $s \le t$. We define the derivatives $|\partial \boldsymbol{y}_t / \partial \boldsymbol{x}_s|$ as the *influential score* to represent the importance of the $s$-th

---

[2]More rigorously, by zero-order hold, $\boldsymbol{b}$ should be parameterized as $\boldsymbol{b} = (\Delta \boldsymbol{A})^{-1}(\exp(-\Delta \boldsymbol{A}) - \boldsymbol{I})\boldsymbol{b}$. However, the presented form is more commonly used in practice as in Gu & Dao (2023).

input token to the $t$-th output token. Note that $|\partial \boldsymbol{y}_t / \partial \boldsymbol{x}_s|$ is well-defined for every $s, t \in [T]$ as long as $(\boldsymbol{A}_t.\boldsymbol{b}_t, c_t, \Delta_t)$ are all differentiable in terms of $\boldsymbol{x}$. Intuitively, if $|\partial \boldsymbol{y}_t / \partial \boldsymbol{x}_s|$ is larger, then the $s$-th input token is more influential on the $t$-th output token, and vice versa.

Below we present a formal result regarding the influential score.

**Theorem 3.1** (Recency of SSMs). *Consider an SSM defined in Eq. 1 with $\{(\boldsymbol{A}_t, \boldsymbol{b}_t, c_t, \Delta_t)\}_{t \in [T]}$. Assume that (i) the input space $\mathcal{X} \subset \mathbb{R}^T$ is compact, (ii) $\{(\boldsymbol{A}_t, \boldsymbol{b}_t, c_t, \Delta_t)\}_{t \in [T]}$ are continuous and have continuous derivatives, and (iii) $\boldsymbol{A}_t \in (0, 1)^{N \times N}$ are diagonal matrices for all $t \in [T]$. Let $A_{max} = \max_{t \in [T], n \in [N]} (\boldsymbol{A}_t)_{n,n}$. Then for arbitrary $\boldsymbol{x} \in \mathcal{X}$ and every $s, t \in [T]$ such that $s < t$, $|\partial \boldsymbol{y}_t / \partial \boldsymbol{x}_s| = O(\exp(-\kappa(t-s)))$ for some $\kappa = \Theta(\log(A_{max}^{-1}))$.*

The proof can be found in Appendix D.1. The first two assumptions are standard and always satisfied. The third assumption also holds for most of SSMs discussed in Sec. 2. Therefore, Theorem 3.1 applies to numerous SSMs including S4 (Gu et al., 2021a; 2022a), Mamba (Gu & Dao, 2023), and many LAMs (Sun et al., 2023; Peng et al., 2023; Yang et al., 2023; Qin et al., 2024; De et al., 2024). Theorem 3.1 states that influential scores between two tokens modeled by SSMs are *exponentially* diminishing with respect to their relative distance. The decay rate is determined by the maximal values among all $\boldsymbol{A}_t$'s elements. The closer $A_{max}$ is to zero, the faster the influential scores decay. The practical implication is that SSMs are factually recency-biased models. Tokens farther away are under-reaching and forgotten rapidly while the information of closer tokens dominates the final output. This can significantly limit their ability of fitting complex long-range relationships.

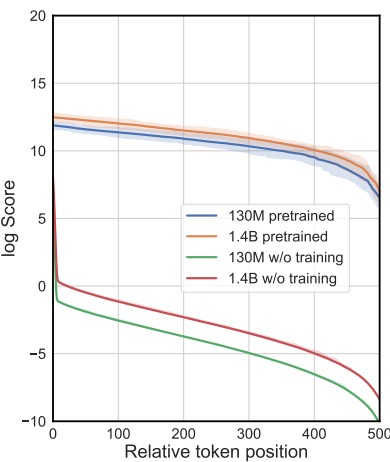

Figure 1: Visualization of log influential scores $\log |\partial \boldsymbol{y}_t / \partial \boldsymbol{x}_s|$ versus distance $(t-s)$.

**Empirical Validation.** In Fig. 1, we plot the logarithmic influence scores against relative distances. Across various model sizes, Mamba consistently exhibits a linear decay rate of influence scores, both at initialization and throughout training. This observation suggests that recency arises from an inherent model bias, as described in Theorem 3.1, rather than solely from capturing data statistics. Moreover, the adopted initialization scheme further amplifies the locality bias.

**Is the design of decay necessary or desirable?** One key observation from our theory is that the parameterization of $\boldsymbol{A}_t$ within the interval $(0, 1)$ leads to strictly decaying dependencies among tokens based on their relative distances. This design choice appears to be a standard practice, and perhaps intentional, in several recently proposed SSMs (Gu & Dao, 2023; Dao & Gu, 2024; Beck et al., 2024; Yang et al., 2023; Peng et al., 2024; De et al., 2024; Ma et al., 2024; Liu et al., 2024; Yang et al., 2024). Interestingly, it has been demonstrated that this "intentional" decay of long-range dependencies not only avoids degrading the perplexity of language models but also improves generalization for length extrapolation. This observation aligns with the empirical success of soft gating mechanisms adopted in traditional RNNs (Cho, 2014; Cho et al., 2014; Gu et al., 2021b) and the decaying patterns imposed on transformers (Raffel et al., 2020; Press et al., 2021; Sun et al., 2022). We find the constraint $\boldsymbol{A}_t \in (0, 1)^{N \times N}$ is likely inherent to SSMs as it plays a critical role in ensuring the numerical stability for length generalization during long-context recurrence (Gu et al., 2022a; Yang et al., 2023). Promoting the importance of local tokens could also lead to a nearly correct bias, as natural language generation mostly utilizes recent contexts. However, as we will detail in subsequent sections, this design comes with a significant drawbacks: it result in substantial loss of long-distance information (Sec. 3.2) and may raise potential security concerns (Sec. 3.3). This observation underscores that the validation perplexity may not fully capture all aspects of a model's capabilities and can be an inadequate metric for assessing long-range dependencies.

## 3.2 LOST IN THE DISTANCE: LONG-CONTEXT RETRIEVAL TEST

To assess the ability of large language models (LLMs) to effectively utilize long-context data, we evaluate open-source SSM using the "Needle in a Haystack" benchmark and compare its performance

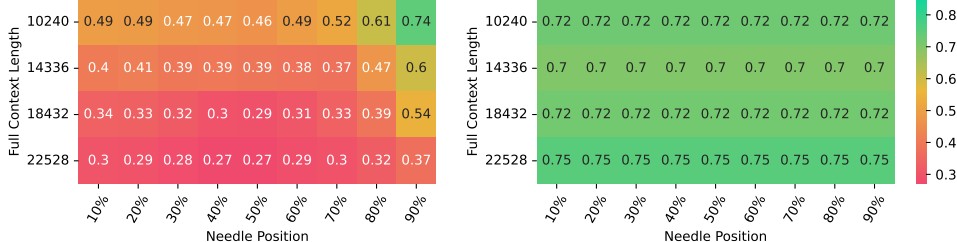

Figure 2: Comparison between SSM and Transformer on the "Needle in a Haystack" benchmark. The left figure shows the retrieval accuracy of the Mamba-Codestral-7B model, while the right figure presents the retrieval accuracy of the Mistral-7B model. We present a heatmap where "full context length" refers to the total length of the document, and "needle position" denotes the relative position of the statement to be retrieved within the context. See more fine-grained visualization in Appendix E.2.

with that of Transformer. In this benchmark, a randomly generated statement is embedded within the middle of a long document, and the models are tasked with retrieving the statement. By varying the insertion position of the statement, we measure the retrieval accuracy at each location, which reflects the model's positional bias. To enforce LLMs using the data within context, instead of recalling information memorized by its model weights, we carefully design the statement with factual error. See detailed examples in Appendix E.2.

We compare the retrieval accuracy of the Mamba-Codestral-7B model, a representative SSM capable of handling long-context inputs of up to 256k tokens, with Mistral-7B (Jiang et al., 2023), which utilizes a transformer architecture. As illustrated in Figure 2, the retrieval accuracy of the Transformer remains stable regardless of the needle position. In contrast, the SSM achieves higher accuracy when the needle is placed closer to the end of the context (i.e., larger needle position values), while its accuracy drops when the needle is located near the beginning of the document. This indicates a positional bias towards local tokens in the SSM.

### 3.3 POTENTIAL RISK ON MODEL ROBUSTNESS

We conduct quantitative experiments to show the recency-biased nature of SSMs will lead to potential hazards. The downstream task in this study is image classification on sequences of pixels (Tay et al., 2020), where $W \times H$ images are flattened to sequences of pixel tokens and fed to sequence models for classification. We test a family of SSMs, including H3 (Fu et al., 2022), RWKV (Peng et al., 2023), and Mamba (Gu & Dao, 2023), and compare them against a transformer baseline (Vaswani et al., 2017) on CIFAR-10 dataset. To adapt SSMs for this task, we append a learnable class token after the last token of the input sequence. The output state of this class token is then mapped to logits using a classifier head. Experiment details are given in Appendix E.3. In the following, two attack patterns on the input data are introduced, which degrade the robustness of SSMs in this task.

**Adversarial Attack.** To assess the bias of SSMs towards corrupted data, we perturb the leading and trailing tokens of input sequences with random noise. In unbiased models, perturbations in both leading and trailing tokens cause similar performance drops. However, in locally biased models, where the class token is appended after the last input token, the trailing tokens are supposed to have greater impacts on classification outcomes than leading tokens. Table 1 presents our experimental results on the CIFAR-10 dataset under two corruption ratios. For each ratio, the same number of

| Models | (no corrupt) | **Corrupted region** (seq. length = 1024) | | | |
|---|---|---|---|---|---|
| | | [992:1024] | [0:32] | [928:1024] | [0:96] |
| H3 | 0.654 | 0.569 (↓ 13.04%) | 0.654 (↓ 0.03%) | 0.477 (↓ 27.07%) | 0.650 (↓ 0.72%) |
| Transformer | 0.580 | 0.535 (↓ 7.81%) | 0.447 (↓ 22.95%) | 0.431 (↓ 25.76%) | 0.370 (↓ 36.32%) |
| RWKV | 0.474 | 0.150 (↓ 68.35%) | 0.466 (↓ 1.58%) | 0.138 (↓ 70.88%) | 0.460 (↓ 2.91%) |
| Mamba | 0.674 | **0.126 (↓ 81.24%)** | 0.658 (↓ 2.30%) | **0.098 (↓ 85.46%)** | 0.647 (↓ 3.98%) |

Table 1: Results of adversarial attack experiments on the CIFAR-10 dataset, evaluated using classification accuracy. Each input sequence contains 1,024 tokens. Two corruption ratios (32/1024 and 96/1024) are applied to perturb the leading and trailing tokens, respectively.

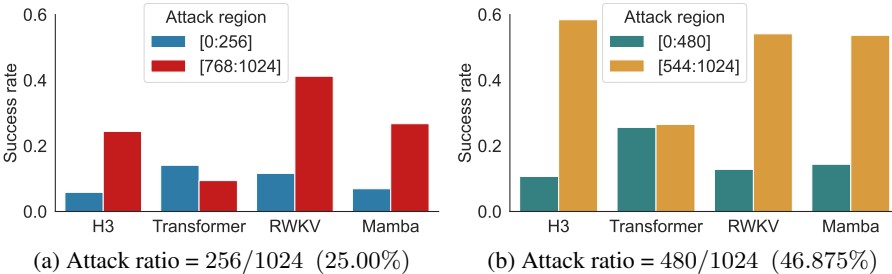

(a) Attack ratio = 256/1024 (25.00%)    (b) Attack ratio = 480/1024 (46.875%)

Figure 3: Results of target attack experiments on CIFAR-10, where "horse" is the target class. (a) and (b) present target attack success rates under two attack ratios. Lower success rates suggest higher robustness in the corresponding attack regions.

leading and trailing tokens are corrupted with Gaussian noise. Among all the SSM family methods compared, the performance drops caused by trailing token corruption are significantly larger than those caused by leading token corruption. Notably, for Mamba, perturbing the last 32 out of 1024 tokens results in an 81.24% drop in classification accuracy, whereas corrupting the first 32 tokens only reduces accuracy by 2.30%. In contrast, the transformer baseline shows relatively smaller impacts from trailing token corruption. Instead, our experiments indicate that more informative features from transformers tend to sink in the leading tokens, aligning with the observations in Xiao et al. (2023).

**Target Attack.** Beyond degrading the performance of SSMs by attacking trailing tokens, we also demonstrate that local bias creates a backdoor for target attacks. In this scenario, a target class is selected, and pixel tokens from that class are used to replace those in images from other classes. The attack succeeds when models mis-classify images from other classes as belonging to the target class. Due to the local bias, trailing tokens are expected to be a more effective attack region for SSMs, leading to a significantly higher attack success rate compared to leading tokens. Fig. 3 shows the success rate comparisons across different attack regions and ratios. When trailing regions are replaced with pixels from the target class, SSMs achieve much higher success rates than when leading regions are attacked. This phenomenon is observed at both 25% and 47% attack ratios. By comparison, the transformer model possesses greater robustness, maintaining similar success rates between attacks on leading and trailing tokens.

**Implications for Language Models.** While our adversarial attack experiments are conducted on image datasets, the findings have broader implications for language models. System prompts, which are typically a group of confidential tokens prepended to user inputs, play a critical role in controlling the behavior of language models and preventing undesirable outputs. However, our targeted attack experiments reveal that SSM-based language models are particularly vulnerable to jailbreak attacks (Perez & Ribeiro, 2022; Zou et al., 2023). This is because SSMs prioritize recent information over past tokens, making it easier to bypass system prompts by appending jailbreak instructions at the end of the input. Moreover, our theoretical analysis suggests that fine-tuning LLMs with instructional datasets or human feedback to enforce adherence to system prompts may not resolve this vulnerability, as the recency bias remains inherent to SSM models regardless of weight configurations.

# 4 UNDERSTANDING SCALABILITY BOTTLENECK OF SSMS

## 4.1 NECESSITY AND LIMITS OF DEPTH SCALING

In Sec. 3.1, we have seen that the dependencies between tokens are exponentially decaying with their relative distances in an SSM layer. Consequently, SSMs resemble localized kernels, similar to those employed in various neural architectures such as Convolutional Neural Networks (CNNs) (LeCun et al., 1998) and Graph Neural Networks (GNNs) (Kipf & Welling, 2016). It is a reasonable postulation that increasing the number of layers can extend the model's receptive field (Goodfellow et al., 2016). We justify this hypothesis via a scaling-up experiment with various context lengths and model architectures.

We pretrain Mamba using causal language modeling with two context lengths, {2048, 8192}. Besides, we fix the number of layers at {16, 24, 32, 48, 64, 72} and vary the hidden dimension. We defer

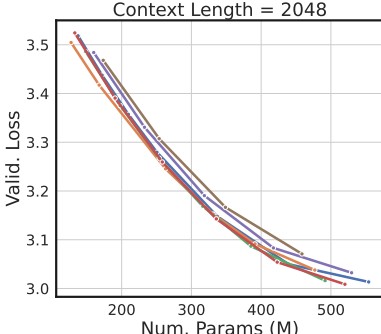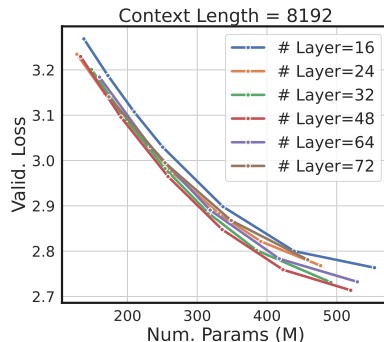

Figure 4: We empirically observe that deeper models become increasingly advantageous as the context length grows. However, beyond a certain depth, the performance of SSMs begins to plateau and eventually declines.

more experiment details in Appendix E.4. The validation loss versus the number of parameters is plotted in Fig. 4. Under the 2048 context length, models of different configurations exhibit similar performance, consistent with the findings of Kaplan et al. (2020). However, as the context length increases, the scaling behavior across depth-width configurations begins to diverge. Notably, deeper models outperform shallower ones, likely because deeper architectures can more effectively utilize the extended context to meet the training objectives. Nevertheless, we observe that the performance gain starts to saturate when we keep increasing the depth (cf. the 32-layer and 48-layer models). When the depth of the model continues to increase, the validation perplexity starts to rise, indicating a decline in performance (cf. the 64-layer and 72-layer models). In Mamba with a 2048 context length, models with more than 48 layers perform worse than 16-layer models. Longer-context models appear to be more tolerant of increased depth, whereas shorter-context models experience a rapid performance degradation once the depth exceeds a certain threshold.

## 4.2 UNVEILING OVER-SMOOTHING IN SSMS

To explain the depth scaling bottleneck revealed in the previous section, we conduct a theoretical and empirical investigation of the feature and state dynamics in SSMs. Our key finding is that token embeddings, after being processed by SSM layers, tend to become increasingly similar, which leads to a phenomenon commonly referred to as *over-smoothing* (NT & Maehara, 2021; Cai & Wang, 2020; Oono & Suzuki, 2019). Over-smoothing occurs when token representations become indistinguishable, rendering the state uninformative.

First of all, we warm up by studying continuous-time S4 with constant $(\boldsymbol{A}, \boldsymbol{b}, \boldsymbol{c})$. Recall that a continuous-time S4 layer can be described by a group of ODEs: $\boldsymbol{h}'(t) = \boldsymbol{A}\boldsymbol{h}(t) + \boldsymbol{b}x(t), \boldsymbol{y}(t) = \boldsymbol{c}^{\top}\boldsymbol{h}(t)$. Our analysis starts with the equivalence between convolution and S4 (Gu et al., 2021b). This is, the analytic solution to the time-invariant ODE can be expressed as $\boldsymbol{y}(t) = \int \boldsymbol{c}^{\top} \exp(\boldsymbol{A}(t - s))\boldsymbol{b}x(s)ds$. Now we analyze the filtering property of this convolution operator from the Fourier domain perspective. We define a convolutional operator as a low-pass filter if it suppresses high-frequency components (see Definition D.3). We summarize the main finding in the following proposition, whose formal version and proof are provided in Appendix D.2.1:

**Proposition 4.1** (Low-pass filtering of continuous S4). *Consider a continuous-time S4 with parameters $(\boldsymbol{A}, \boldsymbol{b}, \boldsymbol{c})$. Assume $\boldsymbol{A}$ is diagonal with all values negative. Then $\boldsymbol{y}(t) = \int \boldsymbol{c}^{\top} \exp(\boldsymbol{A}(t - s))\boldsymbol{b}x(s)ds$ defines a low-pass filter.*

Proposition 4.1 states that S4 is inherently a low-pass filter regardless of how $(\boldsymbol{A}, \boldsymbol{b}, \boldsymbol{c})$ are trained. Therefore, the high-frequency components of input signals are being constantly removed at each layer. Presumably, stacking many S4 layers might cause over-smoothing when all high-frequency components are suppressed to zero.

Now we consider a more general scenario when SSMs work on discrete-time regime and $(\boldsymbol{A}_t, \boldsymbol{b}_t, \boldsymbol{c}_t, \Delta_t)$ are time-varying or even data-dependent. Formally, we prove the following result showing the sharpness of input signals will be reduced as well:

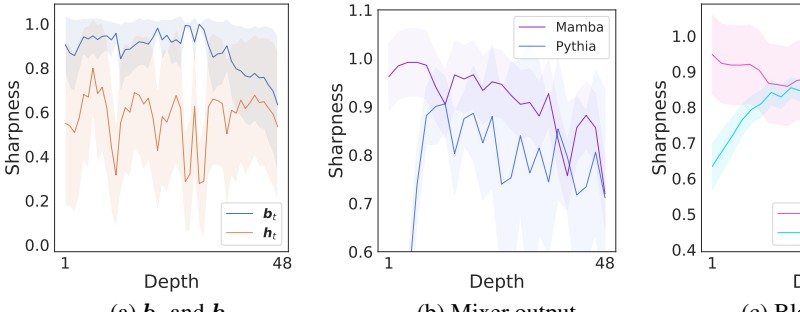

(a) $\boldsymbol{b}_t$ and $\boldsymbol{h}_t$.         (b) Mixer output.         (c) Block output.

Figure 5: Visualization of feature smoothness across layers in pre-trained Mamba and Pythia. The y-axis represents the average pairwise differences among tokens. Mixer outputs (b) solely consider the Mamba or attention module, while Block outputs (c) include all other components (e.g., MLP).

**Theorem 4.2** (Over-smoothing of SSMs). *Consider an SSM specified in Eq. 1 with $\{(\boldsymbol{A}_t, \boldsymbol{b}_t, c_t, \Delta_t)\}_{t \in [T]}$. Assume an input space $\mathcal{X} \subset \mathbb{R}^T$ such that for every $\boldsymbol{x} \in \mathcal{X}$, (i) $(\boldsymbol{A}_t)_{n,n} + \Delta_t \leq 1$ for every $n \in [N]$ and $t \in [T]$, (ii) $\min_{t \in [T]} \boldsymbol{b}_t(\boldsymbol{x}_t)_n \leq 0$ and $\max_{t \in [T]} \boldsymbol{b}_t(\boldsymbol{x}_t)_n \geq 0$ for every $n \in [N]$. Let $A_{min} = \min_{t \in [T], n \in [N]} (\boldsymbol{A}_t)_{n,n}$. Then for any $\boldsymbol{x} \in \mathcal{X}$ and the memory states $\{\boldsymbol{h}_t : t \in [T]\}$ generated by the SSM, we have:*

$$\max_{t,s \in [T]} \|\boldsymbol{h}_t - \boldsymbol{h}_s\|_\infty \leq \left(1 - A_{min}^{T-1}\right) \max_{t,s \in [T]} \|\boldsymbol{b}_t(\boldsymbol{x}_t) - \boldsymbol{b}_s(\boldsymbol{x}_s)\|_\infty, \tag{4}$$

Proof can be found in Appendix D.2.2. We first justify our assumptions here. $(\boldsymbol{A}_t)_{n,n} + \Delta_t \leq 1$ is a generic condition to ensure the recurrence of SSMs is non-expansive, which is crucial to guarantee memory states stay numerically stable. The second assumption requires the data to be well-distributed and centered around the origin, which can be easily satisfied by normalization techniques. We find that prevalent SSM models such as (Gu & Dao, 2023; Peng et al., 2023; De et al., 2024; Qin et al., 2024) can easily achieve these two assumptions.

Moreover, if $(\boldsymbol{A}_t)_{n,n} + \Delta_t = 1$ is always true letting each recurrent update be conservative (Peng et al., 2023; Ma et al., 2022), then we can remove the second assumption as well (see Theorem D.5). Theorem 4.2 examines the relationship between the pairwise distances of memory states and encoded tokens within the sequence. This result indicates that the pairwise discrepancies among memory states are diminished by a factor less than one, suggesting that the memories undergo smoothing following the application of an SSM in Eq. 1. We deem that if the memory is losing its discriminative capacity, the intermediate hidden feature space will similarly collapse.

Delving deeper, the decay rate is intricately linked to both the context length and the minimal value within $\{\boldsymbol{A}_t, t \in [T]\}$. As the context length increases, it requires more time to effectively mix all tokens. This can be understood from the message-passing perspective (Gilmer et al., 2017): the message of the first token needs to go through the whole sequence to be mixed with the last token. When $A_{min}$ approaches one, the decay rate is maximized, as the entire SSM essentially performs a uniform pooling over the entire sequence, which smoothens the signal via a box-like filter. It is worth noting that the smoothing nature of SSMs is intuitive; one can conceptualize the recurrent operation of SSMs as performing a running average of the encoded token signals.

**Empirical Validation.** We adopt a pairwise distance between tokens to quantify the sharpness of a signal: $\mathcal{E}(\boldsymbol{x}) = \frac{1}{2(N-1)} \left(\sum_{i \neq j} \|\boldsymbol{x}_i - \boldsymbol{x}_j\|_2^2\right) / \left(\sum_i \|\boldsymbol{x}_i\|_2^2\right)$. $\mathcal{E}(\boldsymbol{x})$ being small means the token representations are close to each other and become less discriminative. We plot the feature smoothness of a 1.4B Mamba in Fig. 5. In Fig. 5a, $\boldsymbol{b}_t$ is above $\boldsymbol{h}_t$ among all Mamba blocks. This suggests the sharpness of input signals is consistently higher than the sharpness of the memory state output from Mamba, verifying our Theorem 4.2. In addition, Fig. 5b and 5c show the sharpness of Mamba mixer and Mamba block output, which tends to decrease rapidly in deeper layers. We also provide a comparison with a transformer (of the same size). Although transformers suffer from over-smoothing in theory (Dong et al., 2021; Shi et al., 2022; Wang et al., 2022), we observe that transformers have a slower decay of feature sharpness. See a theoretical comparison in Appendix C.

## 5    DISCUSSIONS

In this section, we provide further discussions based on our theory while deferring the remaining parts to Appendix C due to the page limit. We also introduce more related work in Appendix A.

**Revisiting HiPPO theory.**    HiPPO, introduced in (Gu et al., 2020) and extended by Gu et al. (2021b; 2022b), forms the theoretical basis of SSMs. It optimally reconstructs a signal $x$ up to time $t$ by minimizing $\|x_{\leq t} - y^{(t)}\|_{L_2(\omega^{(t)})}$ with respect to a measure $\omega^{(t)}$ supported on $(-\infty, t]$. The solution projects $x_{\leq t}$ onto $N$ basis functions, producing a coefficient vector $\boldsymbol{h}(t)$, which synthesizes $y^{(t)}$ via linear combinations. Gu et al. (2020) showed $\boldsymbol{h}(t)$ evolves as $\boldsymbol{h}'(t) = \boldsymbol{A}(t)\boldsymbol{h}(t) + \boldsymbol{b}(t)x(t)$, where $\omega^{(t)} = \mathbb{I}[0, t]/t$ yields closed form $\boldsymbol{A}(t) = -\boldsymbol{A}_{hippo}/t$. Subsequent works like S4 (Gu et al., 2021a) and Mamba (Gu & Dao, 2023) utilize this matrix as initialization but omitted $1/t$, resulting in a warped measure $\omega^{(t)}(s) \propto \exp(s - t)\mathbb{I}[s < t]$ (Gu et al., 2022b). This measure emphasizes recent history when approximating $x_{\leq t}$. Hence, findings of Gu et al. (2022b) do not contradict our results but also align with Theorem 3.1. The practical implementations often disconnect from HiPPO theory by neglecting the unitary matrices associated with $\boldsymbol{A}_{hippo}$. Whereas, we focus on discrete-domain SSMs, adhering to practical parameterizations. Additionally, Gu et al. (2021b) demonstrates that SSMs' expressiveness spans all convolutions and RNNs. We contend that the low-pass filtering property also arises from the parameter simplifications of $\boldsymbol{A}_t$ (see Proposition 4.1).

**The effect of selection mechanism.**    Traditional S4 architectures operate as linear time-invariant systems. To introduce more non-linearity, Mamba (Gu & Dao, 2023) proposes modeling $(\boldsymbol{b}_t, c_t, \Delta_t)$ as a function of inputs, a mechanism known as *selection*. This is motivated by the selective copying synthetic task, wherein $\boldsymbol{A}_t$ and $\boldsymbol{b}_t$ need to adapt based on content to filter relevant information for memory updates. Despite this adaptation, Theorem 3.1 still holds in scenarios involving the selection mechanism, meaning selective SSMs like Mamba may continue to suffer from recency bias. In the meantime, Theorem 4.2 also applies to Mamba, suggesting that the selective SSMs do not demonstrate higher expressiveness in filtering signals and perform similarly to linear S4 as low-pass filters (Proposition 4.1). Nevertheless, we note that selection can alleviate these issues by adaptively controlling the values in $\boldsymbol{A}_t$. According to our theory, the selection mechanism can potentially make the upper bound $A_{max}$ closer to one and the lower bound $A_{min}$ closer to zero. However, parameter $\boldsymbol{A}$ in Eq. 3 is initialized with negative integers (Gu et al., 2022a), which exacerbates the bound in Theorem 3.1 by accelerating the decay rate of the influence score.

**Complex parameterization.**    Our analysis in the main text primarily focuses on the case where $(\boldsymbol{A}_t, \boldsymbol{b}_t, c_t)$ are both real-valued. While most modern SSMs adopt real parameterizations, complex parameterizations – particularly complex-valued $\boldsymbol{A}_t$ – have been explored in prior works such as Gupta et al. (2022); Goel et al. (2022); Gu et al. (2022a). Notably, we show that both our locality result (Theorem 3.1) and over-smoothing result (Proposition 4.1) remain valid for complex-valued SSMs. We formalize these extensions in Theorems D.2 and D.4. These findings collectively demonstrate that complex parameterization does not eliminate the locality or over-smoothing bias inherent in SSMs.

**Importance of context scaling.**    Theorem 4.2 also highlights the importance of context-length scaling to mitigate the over-smoothing issue. As the smoothing rate decreases with increased context length, lengthening training sequences not only relieves over-smoothing but also maximizes the utility of hardware efficiency of SSMs. This is also evidenced by Fig. 4, where models with longer training contexts have better tolerance of deeper architectures. To enable SSMs to fully utilize the context, increasing the model depth is essential. Consequently, a synchronized scaling of both model depth and context length is required. Investigating the scaling laws governing these two dimensions presents an intriguing direction for future research.

## 6    MITIGATING RECENCY AND OVER-SMOOTHING VIA POLARIZATION

In this section, we propose a simple solution to mitigate recency and over-smoothing simultaneously. First of all, results in Theorem 3.1 and Theorem 4.2 can be interpreted in conjunction. To relieve the smoothening rate, one might aim to minimize the values in $\boldsymbol{A}_t$. However, this could inadvertently enhance the locality of SSMs, as a decrease in $A_{max}$ may occur. A practical implication of this

relationship is that the values in $\boldsymbol{A}_t$ should be as diverse as possible to simultaneously mitigate the artifacts of recency and over-smoothing.

Although $A_{min} \approx 0$ and $A_{max} \approx 1$ could theo-
retically occur simultaneously, our empirical find-
ings show that these values are largely concentrated
within a narrow range. To illustrate this, we visu-
alize the distribution of $(A_{max} - A_{min})$ across dif-
ferent channels in Fig. 6. Each bin represents the
proportion of channels whose memory state satisfies
$(A_{max} - A_{min})$ being smaller than the corresponding
threshold on the x-axis. Notably, over 60% of chan-
nels have $(A_{max} - A_{min})$ values smaller than 0.5,
indicating that most channels cannot simultaneously
achieve $A_{max} \approx 1$ and $A_{min} \approx 0$. Memory repre-
sentations will inevitably undergo either exponential
diminishing or over-smoothing.

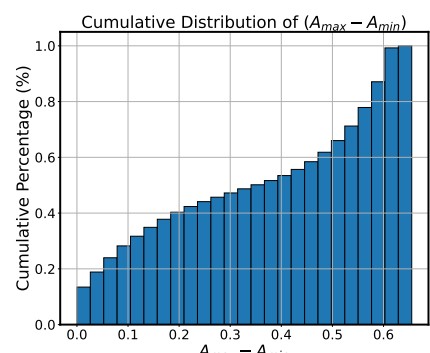

Figure 6: Cumulative histogram of $(A_{max} - A_{min})$. The height of each bin represents the cumulative proportion of $(A_{max} - A_{min})$ less than or equal to the corresponding value on the x-axis.

To this end, we propose to maintain one component
in $\boldsymbol{A}_t$ as a constant 1, another as a constant 0, and
others freely learnable. We term this approach as *polarization*. Polarization ensures that the dimension
polarized to zero in the state memory consistently focuses on the current token, counteracting over-
smoothing by preventing mixing with previous tokens. Simultaneously, another dimension polarized
to one exclusively retains information from past tokens, avoiding locality issues by preserving the
complete history. In our implementation (see Appendix E.5), we polarize the first state channel to
one (*i.e.*, $(\boldsymbol{A}_t)_{1,1} = 1$) and the last state channel to zero (*i.e.*, $(\boldsymbol{A}_t)_{N,N} = 0$).

We empirically validate our polariza-
tion technique through the *associative
recall* tasks, where Mamba models
(Gu & Dao, 2023) are trained over se-
quences of key-value pairs to retrieve
the associated value according to the
query from the context. Please refer
to Arora et al. (2023) and Appendix
E.5 for more details. If an SSM can re-
call information with higher accuracy
from a larger number of key-value
pairs, then it has better long-context
capability. We test the performance of

| Configurations | # Layers | # KV Pairs | | | Avg. |
|---|---|---|---|---|---|
| | | 64 | 128 | 256 | |
| Default $\boldsymbol{A}_t$ | 2 | 98.38 | 81.81 | 36.00 | 72.06 |
| Default $\boldsymbol{A}_t$ | 4 | 99.23 | 82.08 | 33.52 | 71.61 |
| $(\boldsymbol{A}_t)_{1,1} = 1$ | 2 | 99.81 | 94.70 | 56.39 | 83.63 |
| $(\boldsymbol{A}_t)_{N,N} = 0$ | 2 | 98.41 | 81.35 | 36.55 | 72.10 |
| $(\boldsymbol{A}_t)_{N,N} = 0$ | 4 | 99.74 | 92.20 | 52.21 | 81.38 |
| $(\boldsymbol{A}_t)_{1,1} = 1, (\boldsymbol{A}_t)_{N,N} = 0$ | 2 | 99.23 | 95.54 | 54.74 | 83.17 |
| $(\boldsymbol{A}_t)_{1,1} = 1, (\boldsymbol{A}_t)_{N,N} = 0$ | 4 | 99.94 | 98.80 | 81.56 | 93.43 |

Table 2: Results of polarization. Rows 1-2 have no polarization, rows 3-5 only polarize one channel to either one or zero, and rows 6-7 polarize both channels.

Mamba with neither, one of, or both zero- and one-polarized channels, across different numbers
of layers. The empirical results are reported in Tab. 2 (an extended version in Appendix E.5).
The key observation is that the default parameterization of $\boldsymbol{A}_t$ suffers from information retrieval
from long context, and deepening the architecture even harms the performance (potentially due to
over-smoothing). However, once we introduce a channel polarized to one, even shallow Mamba could
perform high-accuracy associative recall (row 3). We can also gain the performance by deepening
Mamba if the over-smoothing issue can be bypassed by adopting the zero-polarized channel (row 5).
Furthermore, if we could apply one- and zero-polarized channels, and in the meanwhile deepening
the architecture, we find it achieves the best performance over all settings (row 7).

## 7 CONCLUSION

In this study, we identify two critical limitations of SSMs. First, we reveal that SSMs exhibit a
pronounced recency bias, which undermines their ability to model long-range dependencies, recall
distant information, and maintain robustness. Second, we find that increasing the depth of SSMs
induces over-smoothing, rendering token representations indistinguishable and hindering further
performance improvements. To address these challenges, we propose a polarization technique that re-
serves two channels in the state transition matrices, assigning one a value of zero and the other a value
of one. Both theoretical analysis and empirical evaluations demonstrate that polarization significantly
enhances the long-range modeling capabilities of SSMs while alleviating over-smoothing.

ACKNOWLEDGMENTS

PW thanks Bo Liu and Songlin Yang for the insightful discussions when preparing this manuscript. Work was done while JZ and PS were interning at UT Austin. PL is supported by NSF awards IIS-2239565 and IIS-2428777 for this project.

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

## A    OTHER RELATED WORK

Despite the empirical success of SSMs in various long-range applications (Lieber et al., 2024; Zhu et al., 2024; Zhang et al., 2024b), their theoretical properties and limitations remain underexplored. Arora et al. (2023) leverages associative recall tasks to theoretically analyze the expressiveness of convolutional SSMs, while advocating for input-dependent kernels. Jelassi et al. (2024) separates the representation capacity of transformers and SSMs via coping tasks. Recent work by Merrill et al. (2024) identifies failure modes of SSMs in state-tracking problems through circuit complexity theory. Hahn (2020) characterize the expressivity of SSMs using linear controlled differential equations, and Ali et al. (2024) reveal structural similarities between selective SSMs and attention mechanisms. While these works provide valuable insights consistent with our findings, none directly examine the long-range modeling capability of SSMs. Ben-Kish et al. (2024) points out that the product of gating matrices (formalized in our Lemma D.1) exhibits locality issues, potentially hindering their ability to model long-range dependencies. However, their analysis lacks a formal justification.

The most relevant prior work to our study on recency bias is perhaps Wang & Xue (2024), in which Their Theorem 3.13 reveals exponentially decaying memory for SSMs. Distinct from their findings, our primary contribution lies in analyzing nonlinearity and input-dependent mechanisms widely adopted in modern SSMs (Gu & Dao, 2023; Yang et al., 2023; Arora et al., 2023). To the best of our knowledge, our work presents the first counterargument showing that even with input-dependent SSMs, effective context filtering may not be achieved. Instead, these mechanisms impose a strict recency bias, inheriting the limitations of linear SSMs as previously highlighted by Wang & Xue (2024). In particular, Wang & Xue (2024) only addresses non-linear activations after performing linear SSMs (*i.e.*, S4), with the state transition matrix $\boldsymbol{A}_t$ being independent of inputs. Our analysis, however, considers both $\boldsymbol{A}_t$ and $\boldsymbol{b}_t$ as input-dependent, aligning with settings in Mamba and other follow-up works. The approach in Wang & Xue (2024) does not naturally extend to this case, as they assume the linearity of the sequence mixer (cf. Eq. 9 in Wang & Xue (2024)). Furthermore, we conduct a finer-grained analysis, quantitatively relating the decay rate to the specific values within the input-dependent gating matrix $\boldsymbol{A}_t$. To our best knowledge, none prior work revealed the smoothening nature of SSM operators.

## B    UNIFIED FORMULATION OF SSMS

Linear Attention Models (LAMs) represent a vast family of architectures evolving rapidly. We present a reformulation of several representative LAMs using the SSM framework described in Eq. 1. This reformulation is expected to extend to models not explicitly covered below, including but not limited to Megalodon (Ma et al., 2022; 2024), Hyena (Poli et al., 2023), HGRN2 (Qin et al., 2024), TTT (Sun et al., 2024), Longhorn (Liu et al., 2024), xLSTM (Beck et al., 2024), and minLSTMs/minGRUs (Feng et al., 2024).

**Linear Attention.**    Attention notoriously suffers from quadratic computation and memory complexities. To address this limitation, Tsai et al. (2019); Katharopoulos et al. (2020) finds that if we express self-attention as a linear dot-product of kernel feature maps, then we can interchange the order of matrix products to achieve a linear time complexity. This finding leads to a series of LAMs. Ignoring the denominator therein, we can formalize linear attention via the recurrence in Eq. 1:

$$\text{(LA)} \quad \boldsymbol{A}_t = \boldsymbol{I}, \quad \boldsymbol{b}_t(\boldsymbol{x}_t) = \boldsymbol{k}(\boldsymbol{x}_t), \quad c_t(\boldsymbol{h}_t) = \boldsymbol{q}(\boldsymbol{x}_t)^\top \boldsymbol{h}_t, \quad \Delta_t = v(\boldsymbol{x}_t),$$

where $\boldsymbol{k}, \boldsymbol{q} : \mathbb{R} \to \mathbb{R}^N$ are the kernel feature maps and $v : \mathbb{R} \to \mathbb{R}$ transforms the input features. Considering multi-channel inputs, $\boldsymbol{k}$, $\boldsymbol{q}$, and $v$ become functions processing vectors, and for each channel, the $\boldsymbol{k}$ and $\boldsymbol{q}$ are shared, while $\Delta_t$ being specified individually. The roles of $\boldsymbol{k}, \boldsymbol{q}, v$ are similar to the key, query, and value transformations in standard transformer (Vaswani et al., 2017). While $\boldsymbol{k}, \boldsymbol{q}, v$ are often chosen as linear mappings, many other options exist. Katharopoulos et al. (2020) adds 1+ELU after linear mapping, Choromanski et al. (2020) leverages orthogonal random Fourier features with hyperbolic cosine as activations, and Zhang et al. (2024a) finds MLP with exponential activation effective.

**Retentive Networks (RetNet).**    RetNet (Sun et al., 2023) is another variant of LAMs, proposed as a successor to transformers given its remarkable performance. Each layer of RetNet consists of a key,

query, and value transformation, akin to linear attention. In addition, it imposes a new decaying term over the past states. We find RetNet can be summarized via our formulation in Eq. 1:

$$\text{(RetNet)} \quad \boldsymbol{A}_t = \gamma \boldsymbol{I}, \quad \boldsymbol{b}_t(\boldsymbol{x}_t) = \boldsymbol{k}(\boldsymbol{x}_t), \quad c_t(\boldsymbol{h}_t) = \boldsymbol{q}(\boldsymbol{x}_t)^\top \boldsymbol{h}_t, \quad \Delta_t = v(\boldsymbol{x}_t),$$

where $\gamma \in (0, 1)$ is a (learnable) scalar, $\boldsymbol{k}, \boldsymbol{q} : \mathbb{R} \to \mathbb{R}^N$, $v : \mathbb{R} \to \mathbb{R}$ are linear functions. Similar to Mamba (Gu & Dao, 2023), RetNet shares $\boldsymbol{b}_t$ and $c_t$ across channels while assigning distinct $\Delta_t$ for each channel when handling multi-channel inputs.

**Gated Linear Attention (GLA).** GLA (Yang et al., 2023) introduces gating mechanism, originally from RNNs (Van Der Westhuizen & Lasenby, 2018), to LAMs. Its computational mechanism can be encompassed by our formulation in Eq. 1:

$$\text{(GLA)} \quad \boldsymbol{A}_t = \text{diag}(\boldsymbol{\alpha}(\boldsymbol{x}_t)), \quad \boldsymbol{b}_t(\boldsymbol{x}_t) = \boldsymbol{k}(\boldsymbol{x}_t), \quad c_t(\boldsymbol{h}_t) = \boldsymbol{q}(\boldsymbol{x}_t)^\top \boldsymbol{h}_t, \quad \Delta_t = v(\boldsymbol{x}_t),$$

where $\boldsymbol{\alpha} : \mathbb{R} \to (0, 1)^N$ converts input to gating logits, $\boldsymbol{k}, \boldsymbol{q} : \mathbb{R} \to \mathbb{R}^N$, $v : \mathbb{R} \to \mathbb{R}$ are linear key, query, value mappings. When the inputs are multi-dimensional, we share $\boldsymbol{\alpha}$ across channels while assigning each channel with a separate $\boldsymbol{k}, \boldsymbol{q}, v$ and extending their input dimension accordingly. Linear attention (Katharopoulos et al., 2020) can be regarded as GLA with constant $\boldsymbol{A}_t$, while RetNet (Sun et al., 2023) can be formulated as GLA with input-independent $\boldsymbol{A}_t$ (Liu et al., 2024).

**RWKV.** RWKV is a series of models that linearize attention computation (Peng et al., 2023; 2024). We focus on RWKV-4 (Peng et al., 2023) and demonstrate it can also be reformulated into the structure of SSMs:

$$\text{(RWKV)} \quad \boldsymbol{A}_t = \frac{\exp(-w)\boldsymbol{I}}{\exp(-w) + \exp(k(\boldsymbol{x}_t))}, \quad \boldsymbol{b}_t(\boldsymbol{x}_t) = \boldsymbol{v}(\boldsymbol{x}_t),$$

$$c_t(\boldsymbol{h}_t) = \boldsymbol{q}(\boldsymbol{x}_t)^\top \boldsymbol{h}_t, \quad \Delta_t = \frac{\exp(k(\boldsymbol{x}_t))}{\exp(-w) + \exp(k(\boldsymbol{x}_t))},$$

where $w \in \mathbb{R}_+$ is a learnable coefficient, $k : \mathbb{R} \to \mathbb{R}$ and $\boldsymbol{q}, \boldsymbol{v} : \mathbb{R} \to \mathbb{R}^N$ are linear mappings. If the inputs are vector-valued, $w$ becomes a vector while $k, \boldsymbol{q}, \boldsymbol{v}$ turns into functions that take vectors as inputs. One important property of RWKV is that $(\boldsymbol{A}_t)_{n,n} + \Delta_t = 1$. Later in RWKV-5 and RWKV-6 (Peng et al., 2024), the normalizer is removed for numerical stability.

**Griffin.** The recurrent unit in Griffin (De et al., 2024) can be re-formulated as a kind of SSMs:

$$\text{(Griffin)} \quad \boldsymbol{A}_t = \text{diag}\left(\boldsymbol{\alpha}(\boldsymbol{x}_t)\right), \quad \boldsymbol{b}_t(\boldsymbol{x}_t) = \text{diag}\left(\boldsymbol{i}(\boldsymbol{x}_t)\right), \quad c_t(\boldsymbol{h}_t) = \boldsymbol{h}_t, \quad \Delta_t = \text{diag}\left(\sqrt{1 - \boldsymbol{\alpha}(\boldsymbol{x}_t)^2}\right),$$

where $\boldsymbol{i}(\boldsymbol{x}_t) = \text{sigmoid}(\boldsymbol{W}_x \boldsymbol{x}_t + \boldsymbol{b}_x)$ is an input gate, $\boldsymbol{\alpha}$ is computed in log-space: $\log \boldsymbol{\alpha}(\boldsymbol{x}_t) = -\xi \, \text{softplus}(\boldsymbol{\Gamma}) \odot \text{sigmoid}(\boldsymbol{W}_a \boldsymbol{x}_t + \boldsymbol{b}_a)$, $\odot$ is Hadamard product, $\xi$ is a constant, and $\boldsymbol{\Gamma}, \boldsymbol{W}_a, \boldsymbol{b}_a, \boldsymbol{W}_x, \boldsymbol{b}_x$ are learnable parameters. In particular, the dimension of $\boldsymbol{h}_t$ in Griffin is equal to the dimension of $\boldsymbol{x}_t$. If we consider single-channel $\boldsymbol{x}_t$, then $\boldsymbol{A}_t, \boldsymbol{b}_t$, and $\Delta_t$ are all scalar-valued.

## C  DEFERRED DISCUSSIONS

**Extended discussion with HiPPO theory.** HiPPO established in (Gu et al., 2020), extended by Gu et al. (2021b; 2022b) is the theoretical foundation of SSMs. Consider a signal $x$ and its reconstruction $y^{(t)}$ up to time $t$. To optimally memorizes the history of $x$ using $y^{(t)}$, HiPPO minimizes $\|x_{\leq t} - y^{(t)}\|_{L_2(\omega^{(t)})}$ w.r.t. a measure $\omega^{(t)}$ supported on $(-\infty, t]$. The solution is to project the history of $x$ before time $t$ onto $N$ basis functions (e.g. Legendre polynomials), which yields a time-continuous coefficient vector $\boldsymbol{h}(t)$, and $y^{(t)}$ can be synthesized by linearly combining the $N$ basis using $\boldsymbol{h}(t)$. Gu et al. (2020) shows that the evolution of $\boldsymbol{h}(t)$ follows an ODE $\boldsymbol{h}'(t) = \boldsymbol{A}(t)\boldsymbol{h}(t) + \boldsymbol{b}(t)x(t)$. In particular, Gu et al. (2020) chooses a uniform measure over the past history $\omega^{(t)} = \mathbb{I}[0, t]/t$, which places no approximation bias over the time horizon in contrast to an earlier work (Voelker et al., 2019). As a result, $\boldsymbol{A}(t)$ in Eq. 1 can be written in a closed form: $\boldsymbol{A}(t) = -\boldsymbol{A}_{hippo}/t$, where $\boldsymbol{A}_{hippo}$ is a time-independent constant called the HiPPO matrix. Its various forms have been used as initialization in subsequent works including S4 (Gu et al., 2021a) and Mamba (Gu & Dao, 2023). While HiPPO theory seems to guarantee the long-rangeness for SSMs, the actual form of $\boldsymbol{A}(t)$ employed in S4 and

Mamba drops the normalizer $1/t$. Gu et al. (2022b) shows that this change causes a warp of measure from uniform to $\omega^{(t)}(s) \propto \exp(s - t)\mathbb{I}(\infty, t]$. We note that this warped measure assigns more importance to recent history, and thus, our Theorem 3.1 does not contradict HiPPO theory but also matches the findings in Gu et al. (2022b). We also point out that when adopting the diagonalized form of $\boldsymbol{A}_{hippo}$ (Gu et al., 2021a; Gupta et al., 2022; Gu et al., 2022a), the unitary matrices decomposed from $\boldsymbol{A}_{hippo}$ is sometimes not applied to $\boldsymbol{b}_t$ and $\boldsymbol{c}_t$, which introduces a disconnect between HiPPO theory and its practical implementation. Our paper directly studies the discrete-domain SSMs and aligns with the parameterization used in practice. Another less-discussed property of SSMs is their approximation power for a broad family of operators. In Gu et al. (2021b), SSMs are shown to possess expressiveness that encompasses both convolutions and RNNs. It is worth noting that the original HiPPO-based SSM is not necessarily a low-pass filter. Rather, it is the successive simplifications in the parameterization of $\boldsymbol{A}_t$ that impart the smoothing characteristics of SSMs (see Proposition 4.1).

**Does hungry hungry hippos help?** The key innovation of Hungry Hungry Hippos (H3) (Fu et al., 2022) lies in the introduction of self-gating connections and locally shifting convolutions to improve in-context recall for state space models (SSMs). This design has quickly become a standard backbone for various SSMs (Gu & Dao, 2023; Beck et al., 2024). However, we question its effectiveness in addressing the local rangeness issue in SSMs. The gating mechanism operates at the token level, which impacts the bound in Theorem 3.1 only by a constant factor. Additionally, the introduced convolutions typically use small kernels, which are insufficient to mitigate the exponentially decaying relevance between tokens. As we empirically show in Fig. 2, while Mamba with H3 performs adequately in associative recall tasks when the state size is sufficiently large, a locality bias begins to emerge as the number of key-value pairs exceeds the model's memory capacity. This highlights the limitations of the architecture in handling long-range dependencies under constrained memory.

**Does gradient vanish in SSMs?** Vanishing gradients refer to a challenge in RNNs, where backpropagation-based learning is impeded due to gradient magnitudes decaying exponentially over time. The diminishing dependencies among distant tokens are a fundamental cause of this issue (Bengio et al., 1994). SSMs were initially proposed to address this limitation by explicitly modeling long-range dependencies, as highlighted in (Gu et al., 2020; 2021b). Subsequent work, such as Mamba, extends this approach by adopting their initialization alongside newly proposed selection mechanisms, which are widely believed to enhance these capabilities. Our Theorem 3.1 lies in theoretically challenging this assumption. We demonstrate that modern SSMs still suffer from the recency bias, which not only undermines their ability to capture long-term dependencies but also potentially exacerbates the vanishing gradient problem.

**Connection with over-squashing theory in GNNs.** We can compare recency of SSMs to over-squashing in GNNs. The influential score defined in Sec. 3.1 is also used for *over-squashing* analysis in Graph Neural Networks (GNNs) to identify information bottleneck (Topping et al., 2021; Di Giovanni et al., 2023). The sensitivity analysis in Topping et al. (2021) demonstrates similar exponentially decaying dependencies among graph nodes, dependent on the underlying graph topology. We postulate that propagating information from long distances remains challenging for SSMs because the model needs to encapsulate all history information into a fixed-dimension hidden vector, which is also observed as one major problem with RNNs (Bengio et al., 1994; Alon & Yahav, 2020; Sutskever et al., 2014; Cho et al., 2014; Cho, 2014).

**Connection with over-smoothing theory in GNNs and transformers.** Over-smoothing issues were first identified in GNNs (Li et al., 2018; NT & Maehara, 2021; Oono & Suzuki, 2019; Cai & Wang, 2020; Wu et al., 2022; 2024b) and later explored in transformers (Dong et al., 2021; Wang et al., 2022; Shi et al., 2022; Wu et al., 2024a; Geshkovski et al., 2023). In both cases, over-smoothing manifests as feature representations becoming increasingly uniform with greater model depth. To the best of our knowledge, our work is the first to uncover this phenomenon in SSMs. In GNNs, the over-smoothing effect typically follows a linear decay rate governed by the second-largest eigenvalue of the adjacency matrix: $O(\lambda_2^L)$, where $L$ denotes the number of layers (Oono & Suzuki, 2019; Cai & Wang, 2020). More closely related to our setting, Wu et al. (2024a) analyzes the convergence rate of feature smoothness in causal attention, demonstrating a rate of $O((1 - \hat{A}_{min}^T)^{L/T})$, where $\hat{A}_{min}$ is the minimal value in attention maps. In comparison, our result in Theorem 4.2 shows a rate of $O((1 - A_{min}^T)^L)$, indicating that the smoothening speed of SSMs is

faster than that of transformers. Our Fig. 5 supports this claim, as the transformer-based architecture exhibits a mild decrease of sharpness at deeper layers, whereas SSMs show a constantly steeper decay slope. Moreover, due to the inherent locality of SSMs (Theorem 3.1), achieving effective long-range interactions necessitates deeper architectures. In contrast, transformers, which allow arbitrary long-range interactions among tokens, do not have the same requirement. This distinction is partially supported by the observation that modern SSMs often adopt architectures that are roughly twice as deep as transformers. Consequently, depth-scaling limitations are more critical for SSMs than for transformers.

# D PROOFS

## D.1 EXPONENTIALLY DECAYING DEPENDENCY WITH RELATIVE DISTANCE

In this section, we extend and prove Theorem 3.1 in a more general case where inputs and parameters are all complex-valued. Below we by default assume that $A_t \in \mathbb{C}^{N \times N}$, $b_t : \mathbb{C} \to \mathbb{C}^N$, $c_t : \mathbb{C}^N \to \mathbb{C}$, and $\Delta_t \in \mathbb{R}$. First of all, we present the following auxiliary lemma, reformulating SSM recurrence in an explicit parallel form.

**Lemma D.1** (Parallel form). *For any $\{(A_t, b_t, c_t, \Delta_t)\}_{t \in [T]}$ and $x \in \mathbb{C}^T$, $y \in \mathbb{C}^T$ computed via an SSM defined in Eq. 1 is equal to:*

$$h_t = \sum_{s=1}^{t-1} \left( \prod_{r=s+1}^{t} A_r \right) \Delta_s b_s(x_s) + \Delta_t b_t(x_t), \quad y_t = c_t(h_t), \quad \forall t \in [T]. \tag{5}$$

*Proof.* Proof by induction. First let us examine the base case when $t = 1$: $h_1 = \Delta_1 b_1(x_1)$, indicating that Eq. 5 holds trivially. Now we make inductive hypothesis that Eq. 5 holds for some $t \in [T-1]$. Then for time step $t + 1 \in [T]$:

$$h_{t+1} = A_{t+1} h + \Delta_{t+1} b_{t+1}(x_{t+1})$$

$$= A_{t+1} \left( \sum_{s=1}^{t-1} \left( \prod_{r=s+1}^{t} A_r \right) \Delta_s b_s(x_s) + \Delta_t b_t(x_t) \right) + \Delta_{t+1} b_{t+1}(x_{t+1})$$

$$= \sum_{s=1}^{t-1} \left( A_{t+1} \cdot \prod_{r=s+1}^{t} A_r \right) \Delta_s b_s(x_s) + A_{t+1} \Delta_t b_t(x_t) + \Delta_{t+1} b_{t+1}(x_{t+1})$$

$$= \sum_{s=1}^{t-1} \left( \prod_{r=s+1}^{t+1} A_r \right) \Delta_s b_s(x_s) + \left( \prod_{r=t+1}^{t+1} A_r \right) \Delta_t b_t(x_t) + \Delta_{t+1} b_{t+1}(x_{t+1})$$

$$= \sum_{s=1}^{t} \left( \prod_{r=s+1}^{t+1} A_r \right) \Delta_s b_s(x_s) + \Delta_{t+1} b_{t+1}(x_{t+1}),$$

which also satisfies Eq. 5. Then we conclude the proof by induction. $\square$

**A remark of Lemma D.1.** Lemma D.1 provides an alternative perspective on how SSMs compute the outputs. The predicted value for the $t$-th token is obtained via decoding a weighted aggregation over representations of all past tokens. The encoding and decoding stage is element-wise independent of the context. Whereas, the "weight" associated with each past token in the summation reflects the pairwise relationship, playing a similar role to attention weights in transformers (Dao & Gu, 2024; Ali et al., 2024). The weight corresponding to one past token is calculated as the cumulative product $\prod_r A_r$, where $r \in [s+1, t]$ traverses from the past token (at time $s$) to the target token (at time $t$). Assume $A_t \in (0, 1)^{N \times N}$, which is satisfied by most of SSMs discussed in Sec. 2, we can show that $(\prod_{r=s+1}^{t} A_r)_{n,n} < (\prod_{r=s'+1}^{s} A_r)_{n,n}$ for any $s < s' < t$ and $n \in [N]$. By this interpretation, SSMs assign strictly higher "attention" to the nearer tokens than the further tokens.

Below we formally state and prove the complex version of Theorem 3.1. Theorem 3.1 is a straightforward corollary of Theorem D.2.

**Theorem D.2** (Recency of SSMs). *Consider an SSM defined in Eq. 1 with $\{(\boldsymbol{A}_t, \boldsymbol{b}_t, c_t, \Delta_t)\}_{t \in [T]}$. Assume that:*

    *(i) The input space $\mathcal{X} \subset \mathbb{C}^T$ is compact.*

    *(ii) $\{(\boldsymbol{A}_t, \boldsymbol{b}_t, c_t, \Delta_t)\}_{t \in [T]}$ and $\left\{ \left( \frac{\partial \boldsymbol{A}_t}{\partial \boldsymbol{x}_t}, \frac{\partial \boldsymbol{b}_t}{\partial \boldsymbol{x}_t}, \frac{\partial c_t}{\partial \boldsymbol{x}_t}, \frac{\partial \Delta_t}{\partial \boldsymbol{x}_t} \right) \right\}_{t \in [T]}$ are continuous.*

    *(iii) $\boldsymbol{A}_t$ is diagonal and $0 < |(\boldsymbol{A}_t)_{n,n}| < 1$ for all $t \in [T], n \in [N]$.*

*Let $A_{max} = \max_{t \in [T], n \in [N]} |(\boldsymbol{A}_t)_{n,n}|$. Then for arbitrary $\boldsymbol{x} \in \mathcal{X}$ and every $s, t \in [T]$ such that $s < t$,*

$$\left| \frac{\partial \boldsymbol{y}_t}{\partial \boldsymbol{x}_s} \right| \leq C \exp\left(-\kappa(t - s)\right),$$

*for some constant $C > 0$ independent of $t$ and $s$, and $\kappa = \log(A_{max}^{-1})$.*

*Proof of Theorem 3.1.* First of all, we note that by compactness of input space (Assumption (i)) and continuity (Assumption (ii)), $(\boldsymbol{A}_t, \boldsymbol{b}_t, c_t, \Delta_t)$ and $\left( \frac{\partial \boldsymbol{A}_t}{\partial \boldsymbol{x}_t}, \frac{\partial \boldsymbol{b}_t}{\partial \boldsymbol{x}_t}, \frac{\partial c_t}{\partial \boldsymbol{x}_t}, \frac{\partial \Delta_t}{\partial \boldsymbol{x}_t} \right)$ are all bounded by some constant for every $t \in [T]$.

By Lemma D.1, Eq. 1 can be expressed in the closed form as follows:

$$
\begin{aligned}
\boldsymbol{h}_t &= \sum_{i=1}^{t-1} \left( \prod_{r=i+1}^{t} \boldsymbol{A}_r \right) \Delta_i \boldsymbol{b}_i(\boldsymbol{x}_i) + \Delta_t \boldsymbol{b}_t(\boldsymbol{x}_t) \\
&= \sum_{i=1}^{t-1} \exp \left( \sum_{r=i+1}^{t} \log \boldsymbol{A}_r \right) \Delta_i \boldsymbol{b}_i(\boldsymbol{x}_i) + \Delta_t \boldsymbol{b}_t(\boldsymbol{x}_t) \\
&= \underbrace{\sum_{i=1}^{s-1} \exp \left( \sum_{r=i+1}^{t} \log \boldsymbol{A}_r \right) \Delta_i \boldsymbol{b}_i(\boldsymbol{x}_i)}_{\boldsymbol{u}_t} + \underbrace{\sum_{i=s}^{t-1} \exp \left( \sum_{r=i+1}^{t} \log \boldsymbol{A}_r \right) \Delta_i \boldsymbol{b}_i(\boldsymbol{x}_i) + \Delta_t \boldsymbol{b}_t(\boldsymbol{x}_t)}_{\boldsymbol{v}_t},
\end{aligned}
$$

where the second equality is by simply rewriting the cumulative product as an exponential of cumulative summation of logarithms. The logarithmic terms are well defined and constantly negative due to Assumption (iii). We decompose $\boldsymbol{h}_t$ into three components, where the first term, denoted as $\boldsymbol{u}_t$, depends on $\boldsymbol{x}_s$ only through $\{\boldsymbol{A}_r\}_{r=s}^t$, the second term, denoted as $\boldsymbol{v}_t$, only relies on $\boldsymbol{x}_s$ via $\{\Delta_i \boldsymbol{b}_i\}_{i=s}^t$, while the remaining part is independent of $\boldsymbol{x}_s$.

Now we tackle each component separately. First, we simplify $\partial \boldsymbol{u}_t / \partial \boldsymbol{x}_s$ as:

$$
\begin{aligned}
\frac{\partial \boldsymbol{u}_t}{\partial \boldsymbol{x}_s} &= \frac{\partial}{\partial \boldsymbol{x}_s} \left[ \sum_{i=1}^{s-1} \exp \left( \sum_{r=i+1}^{t} \log \boldsymbol{A}_r \right) \Delta_i \boldsymbol{b}_i(\boldsymbol{x}_i) \right] \\
&= \sum_{i=1}^{s-1} \frac{\partial}{\partial \boldsymbol{x}_s} \left( \exp \left( \sum_{r=i+1}^{t} \log \boldsymbol{A}_r \right) \right) \Delta_i \boldsymbol{b}_i(\boldsymbol{x}_i) \\
&= \sum_{i=1}^{s-1} \exp \left( \sum_{r=i+1}^{t} \log \boldsymbol{A}_r \right) \left( \sum_{r=i+1}^{t} \frac{\partial \log \boldsymbol{A}_r}{\partial \boldsymbol{x}_s} \right) \Delta_i \boldsymbol{b}_i(\boldsymbol{x}_i) \\
&= \sum_{i=1}^{s-1} \exp \left( \sum_{r=i+1}^{t} \log \boldsymbol{A}_r \right) \left( \boldsymbol{A}_s^{-1} \frac{\partial \boldsymbol{A}_s}{\partial \boldsymbol{x}_s} \right) \Delta_i \boldsymbol{b}_i(\boldsymbol{x}_i),
\end{aligned}
$$

where the last steps follow from the basic chain rule. Then we can bound its $\ell_1$ norm by:

$$\left\| \frac{\partial \boldsymbol{u}_t}{\partial \boldsymbol{x}_s} \right\|_1 \leq \sum_{n=1}^{N} \sum_{i=1}^{s-1} \left| \exp\left( \sum_{r=i+1}^{t} \log \boldsymbol{A}_r \right)_{n,n} \right| \cdot \left| \left( \frac{1}{\boldsymbol{A}_r} \frac{\partial \boldsymbol{A}_r}{\partial \boldsymbol{x}_s} \right)_{n,n} \Delta_i \boldsymbol{b}_i(\boldsymbol{x}_i)_n \right| \tag{6}$$

$$\leq C_1 \sum_{n=1}^{N} \sum_{i=1}^{s-1} \exp\left( \sum_{r=i+1}^{t} \log|(\boldsymbol{A}_r)_{n,n}| \right) \tag{7}$$

$$\leq C_1 N \sum_{i=1}^{s-1} \exp\left( \log A_{max}(t-i) \right) \tag{8}$$

$$= C_1 N \frac{1 - \exp\left( -\log A_{max}(s-1) \right)}{1 - A_{max}^{-1}} \exp\left( \log A_{max}(t-1) \right) \tag{9}$$

$$= C_1 N \frac{\exp\left( \log A_{max}(t-s) \right) - \exp\left( \log A_{max}(t-1) \right)}{A_{max}^{-1} - 1}$$

$$\leq \frac{C_1 N}{A_{max}^{-1} - 1} \exp\left( -\log A_{max}^{-1}(t-s) \right). \tag{10}$$

We elaborate on the derivation step by step. Eq. 6 is due to triangular inequality. To obtain Eq. 7, we note that $\boldsymbol{A}_r^{-1}, \partial \boldsymbol{A}_r / \partial \boldsymbol{x}_s, \Delta_i, \boldsymbol{b}_i$ are all element-wisely bounded, thus, we can extract their uniform upper bound $C_1 > 0$ out of the summation Eq. 8 can be derived by applying the supremum $A_{max}$ over all $\{|(\boldsymbol{A}_t)_{n,n}|\}_{t\in[T],n\in[N]}$. Eq. 9 follows from the summation of geometric series. And finally, inequality in Eq. 10 holds by dropping a negative term.

Similarly, we rewrite $\partial \boldsymbol{v}_t / \partial \boldsymbol{x}_s$ as below:

$$\frac{\partial \boldsymbol{v}_t}{\partial \boldsymbol{x}_s} = \frac{\partial}{\partial \boldsymbol{x}_s} \left[ \sum_{i=s}^{t-1} \exp\left( \sum_{r=i+1}^{t} \log \boldsymbol{A}_r \right) \Delta_i \boldsymbol{b}_i(\boldsymbol{x}_i) \right]$$

$$= \sum_{i=s}^{t-1} \exp\left( \sum_{r=i+1}^{t} \log \boldsymbol{A}_r \right) \frac{\partial(\Delta_i \boldsymbol{b}_i(\boldsymbol{x}_i))}{\partial \boldsymbol{x}_s}$$

$$= \exp\left( \sum_{r=s+1}^{t} \log \boldsymbol{A}_r \right) \frac{\partial(\Delta_s \boldsymbol{b}_s(\boldsymbol{x}_s))}{\partial \boldsymbol{x}_s},$$

by which we can yield the following upper bound on its $\ell_1$ norm:

$$\left\| \frac{\partial \boldsymbol{v}_t}{\partial \boldsymbol{x}_s} \right\|_1 \leq \sum_{n=1}^{N} \left| \exp\left( \sum_{r=s+1}^{t} \log \boldsymbol{A}_r \right)_{n,n} \right| \cdot \left| \frac{\partial(\Delta_s \boldsymbol{b}_s(\boldsymbol{x}_s)_n)}{\partial \boldsymbol{x}_s} \right|$$

$$\leq C_2 \sum_{n=1}^{N} \exp\left( \sum_{r=s+1}^{t} \log|(\boldsymbol{A}_r)_{n,n}| \right) \tag{11}$$

$$\leq N C_2 \exp\left( -\log A_{max}^{-1}(t-s) \right), \tag{12}$$

where Eq. 11 is obtained by applying uniform upper bound $C_2 > 0$ over $|\partial(\Delta_s \boldsymbol{b}_s(\boldsymbol{x}_s)_n)/\partial \boldsymbol{x}_s|$. Eq. 12 is induced by leveraging the supremum $A_{max}$ over all $\{|(\boldsymbol{A}_t)_{n,n}|\}_{t\in[T],n\in[N]}$.

Combining Eqs. 10 and 12, we have:

$$\left\| \frac{\partial \boldsymbol{h}_t}{\partial \boldsymbol{x}_s} \right\|_1 \leq \left( \frac{C_1 N}{A_{max}^{-1} - 1} + N C_2 \right) \exp\left( -\log A_{max}^{-1}(t-s) \right). \tag{13}$$

Finally, we conclude the proof based on Eq. 13:

$$\left| \frac{\partial \boldsymbol{y}_t}{\partial \boldsymbol{x}_s} \right| = \left| \frac{\partial c_t(\boldsymbol{h}_t)}{\partial \boldsymbol{h}_t}^{\top} \frac{\partial \boldsymbol{h}_t}{\partial \boldsymbol{x}_s} \right| \leq \left\| \frac{\partial c_t(\boldsymbol{h}_t)}{\partial \boldsymbol{h}_t} \right\|_{\infty} \left\| \frac{\partial \boldsymbol{h}_t}{\partial \boldsymbol{x}_s} \right\|_1$$

$$\leq C_3 \left( \frac{C_1 N}{A_{max}^{-1} - 1} + N C_2 \right) \exp\left( -\log A_{max}^{-1}(t-s) \right),$$

$$= C \exp\left( -\kappa(t-s) \right)$$

where we use Hölder inequality in the first equation and upper bound $\|\partial c_t(\boldsymbol{h}_t)/\partial \boldsymbol{h}_t\|_\infty$ again via constant $C_3 > 0$ due to continuity ($c_t(\boldsymbol{h}_t)$ is a composition of a series of continuous functions as in Assumption (ii)). This is as desired after letting $C > 0$ absorb all constants and $\kappa = \log A_{max}^{-1}$. $\qquad \square$

## D.2 Over-smoothing in SSMs

### D.2.1 Low-pass Filtering Properties

In this section, we formally state and prove Proposition 4.1. To begin with, we define low-pass filters as below:

**Definition D.3** (Low-pass Filter). *Suppose $z(t) : \mathbb{C} \to \mathbb{C}$ is absolutely integrable. Then the Fourier transform $Z(\omega) : \mathbb{C} \to \mathbb{C}$ exists and $Z(\omega) \triangleq \int z(t) \exp(-\mathrm{i}\omega t) dt$. We say $z(t)$ is an $(\epsilon, \Omega)$-low-pass filter for some $0 < \epsilon < 1$ and $\Omega > 0$ if $|Z(\omega)| \leq \epsilon$ for every $|\omega| \geq \Omega$. Furthermore, we say $z(t)$ is a low-pass filter if for every $0 < \epsilon < 1$, there exists $\Omega > 0$ such that $z(t)$ is an $(\epsilon, \Omega)$-low-pass filter.*

Note that our definition of low-pass filters focuses on its effect on removing high-frequency components. By Definition D.3, it is equivalent to say a system performs low-pass filtering if and only if the responses converge to zero as the frequency increases. Fixing $\epsilon > 0$ small enough, we can say a $(\epsilon, \Omega)$-low-pass filter has cut-off frequency at $|\omega| = \Omega$.

Now we present a formal version of Proposition 4.1 as below, which quantifies the cut-off frequency of a filter induced by continuous-time S4. Note that Proposition D.4 holds for complex-valued $(\boldsymbol{A}, \boldsymbol{b}, \boldsymbol{c})$. Below we assume $\boldsymbol{A} \in \mathbb{C}^{N \times N}$, $\boldsymbol{b} \in \mathbb{C}^N$, $\boldsymbol{c} \in \mathbb{C}^N$ by default.

**Proposition D.4** (Formal version of Proposition 4.1). *Consider a continuous-time S4 with parameters $(\boldsymbol{A}, \boldsymbol{b}, \boldsymbol{c})$. Assume $\boldsymbol{A} \in \mathbb{C}^{N \times N}$ is diagonal and its diagonal values have negative real parts. Then $\boldsymbol{y}(t) = \int \boldsymbol{c}^\top \exp(\boldsymbol{A}(t - s))\boldsymbol{b}x(s)ds$ is an $(\epsilon, O(1/\epsilon))$-low-pass filter for any $\epsilon > 0$ sufficiently small.*

*Proof.* First of all, let us rewrite the filter induced by S4 as below:

$$z(t) = \sum_{n=1}^N \boldsymbol{c}_n \boldsymbol{b}_n \exp(\boldsymbol{A}_{n,n} t).$$

Then we apply Fourier transform on $z(t)$:

$$Z(\omega) = \int z(t) e^{-\mathrm{i}\omega t} dt = \int \sum_{n=1}^N \boldsymbol{c}_n \boldsymbol{b}_n \exp(\boldsymbol{A}_{n,n} t) \exp(-\mathrm{i}\omega t) dt$$

$$= \sum_{n=1}^N \boldsymbol{c}_n \boldsymbol{b}_n \int \exp(\boldsymbol{A}_{n,n} t) \exp(-\mathrm{i}\omega t) dt$$

$$= \sum_{n=1}^N \boldsymbol{c}_n \boldsymbol{b}_n \int \exp((\boldsymbol{A}_{n,n} - \mathrm{i}\omega) t) dt$$

$$= \sum_{n=1}^N \frac{\boldsymbol{c}_n \boldsymbol{b}_n}{\mathrm{i}\omega - \boldsymbol{A}_{n,n}},$$

where the last integral converges due to $\Re(\boldsymbol{A}_{n,n}) < 0$. Then we can upper bound the magnitude of $Z(\omega)$ as:

$$|Z(\omega)| = \left| \sum_{n=1}^N \frac{\boldsymbol{c}_n \boldsymbol{b}_n}{\mathrm{i}\omega - \boldsymbol{A}_{n,n}} \right| \leq \sum_{n=1}^N \left| \frac{\boldsymbol{c}_n \boldsymbol{b}_n}{\mathrm{i}\omega - \boldsymbol{A}_{n,n}} \right| \leq \sum_{n=1}^N \frac{|\boldsymbol{c}_n \boldsymbol{b}_n|}{||\omega| - |\boldsymbol{A}_{n,n}||}.$$

We consider $\epsilon > 0$ small enough, thus, it is sufficient to consider the scenario when $|\omega| > A_{max} \triangleq \max_{n \in [N]} |\boldsymbol{A}_{n,n}|$. In this regime, we have:

$$|Z(\omega)| \leq \left( \sum_{n=1}^N |\boldsymbol{c}_n \boldsymbol{b}_n| \right) \frac{1}{|\omega| - A_{max}} \leq \frac{\|\boldsymbol{b}\|_2 \|\boldsymbol{c}\|_2}{|\omega| - A_{max}},$$

where the last two inequalities are due to Hölder's inequality. It is sufficient to set:

$$\Omega = \frac{\|\boldsymbol{b}\|_2 \|\boldsymbol{c}\|_2}{\epsilon} + A_{max},$$

to let $|Z(\omega)| \leq \epsilon$ for every $|\omega| \geq \Omega$ hold. $\qquad\square$

### D.2.2 GENERALIZED SMOOTHENING PROPERTIES

In this section, we define generalized smoothening operators as those that narrow distances between token features. We provide the following results that formalize this point. Theorem D.5 below extends the result of Theorem 4.2 to include the case when $(\boldsymbol{A}_t)_{n,n} + \Delta_t = 1$ for every $n \in [N], t \in [T]$, which is more aligned with Ma et al. (2022; 2024); Peng et al. (2023) in practice.

**Theorem D.5** (Complete version of Theorem 4.2). *Consider an SSM specified in Eq. 1 with $\{(\boldsymbol{A}_t, \boldsymbol{b}_t, c_t, \Delta_t)\}_{t \in [T]}$. Assume an input space $\mathcal{X} \subset \mathbb{R}^T$ such that for every $\boldsymbol{x} \in \mathcal{X}$, either of the following conditions is satisfied:*

- *(i) $(\boldsymbol{A}_t)_{n,n} + \Delta_t = 1$ for every $n \in [N]$ and $t \in [T]$, or*

- *(ii) $(\boldsymbol{A}_t)_{n,n} + \Delta_t \leq 1$ for every $n \in [N], t \in [T]$, and $\min_{t \in [T]} \boldsymbol{b}_t(\boldsymbol{x}_t)_n \leq 0$ and $\max_{t \in [T]} \boldsymbol{b}_t(\boldsymbol{x}_t)_n \geq 0$ for every $n \in [N]$.*

*Let $A_{min} = \min_{t \in [T], n \in [N]}(\boldsymbol{A}_t)_{n,n}$. Then for any $\boldsymbol{x} \in \mathcal{X}$ and the memory states $\{\boldsymbol{h}_t : t \in [T]\}$ generated by the SSM, we have:*

$$\max_{t,s \in [T]} \|\boldsymbol{h}_t - \boldsymbol{h}_s\|_\infty \leq \left(1 - A_{min}^{T-1}\right) \max_{t,s \in [T]} \|\boldsymbol{b}_t(\boldsymbol{x}_t) - \boldsymbol{b}_s(\boldsymbol{x}_s)\|_\infty,$$

*Proof.* To simplify the proof, we first only consider the dynamics of one channel in the memory state. We denote $\alpha_t = (\boldsymbol{A}_t)_{n,n}$, $z_t = \boldsymbol{b}_t(\boldsymbol{x}_t)_n$, and $s_t = \boldsymbol{h}_{t,n}$ for some $n \in [N]$. Additionally, we define the following two quantities:

$$m = \min_{t \in [T]} z_t, \quad M = \max_{t \in [T]} z_t$$

When Assumption (ii) holds, we know that $m \leq 0$ and $M \geq 0$. Suppose $z_1 = pm + (1-p)M$ for some $p \in [0, 1]$. Furthermore, let $q = 1 - p$. Now we consider the following dynamics with $s_0 = 0$:

$$s_t = \alpha_t s_{t-1} + \Delta_t z_t, \quad t \in [T], \tag{14}$$

Next, we can show the result below (proved later):

**Lemma D.6.** *We have the following inequality for Eq. 14:*

$$(1 - A_{min}^{t-1}q)m + A_{min}^{t-1}qM \leq s_t \leq A_{min}^{t-1}pm + (1 - A_{min}^{t-1}p)M, \quad \forall t \in [T],$$

*if either of the following conditions holds:*

- *(i) $\alpha_t + \Delta_t = 1$;*

- *(ii) $\alpha_t + \Delta_t < 1$ and $m \leq 0 \leq M$.*

Note that the two conditions in Lemma D.6 corresponds to Assumptions (i) and (ii) in Theorem D.5, respectively. Thus we can upper and lower bound the minimum and maximum of memory states $\{s_t : t \in [T]\}$ through:

$$m' \triangleq \min_{t \in [T]} s_t \geq \min_{t \in [T]} (1 - A_{min}^{t-1}q)m + A_{min}^{t-1}qM$$
$$= (1 - A_{min}^{T-1}q)m + A_{min}^{T-1}qM,$$

as we are moving the convex combination towards the smaller end (i.e., $1 - A_{min}^{T-1}q > 1 - A_{min}^{t-1}q$), and similarly,

$$M' \triangleq \max_{t \in [T]} s_t \leq \max_{t \in [T]} A_{min}^{t-1}pm + (1 - A_{min}^{t-1}p)M$$
$$\leq A_{min}^{T-1}pm + (1 - A_{min}^{T-1}p)M.$$

as we are now relaxing the convex combination towards the larger end (i.e., $1 - A_{min}^{T-1}p > 1 - A_{min}^{t-1}p$)
Henceforth, we can upper bound:

$$M' - m' \leq A_{min}^{T-1}pm + (1 - A_{min}^{T-1}p)M - (1 - A_{min}^{n-1}q)m - A_{min}^{n-1}qM$$
$$= (1 - A_{min}^{T-1})(M - m),$$

using the fact that $p + q = 1$.

Now we can apply the above result to all memory channels. Assigning each channel with $m_n = \min_{t \in [T]} \boldsymbol{b}_t(\boldsymbol{x}_t)_n$, $M_n = \max_{t \in [T]} \boldsymbol{b}_t(\boldsymbol{x}_t)_n$ and $M'_n, m'_n$ accordingly, where $n \in [N]$. We can yield:

$$\max_{t,s \in [T]} \|\boldsymbol{h}_t - \boldsymbol{h}_s\|_\infty \leq \max_{n \in [N]} (M'_n - m'_n)$$
$$\leq \max_{n \in [N]} \left(1 - A_{min}^{T-1}\right)(M_n - m_n)$$
$$\leq \left(1 - A_{min}^{T-1}\right) \max_{n \in [N]} (M_n - m_n)$$
$$= \left(1 - A_{min}^{T-1}\right) \max_{n \in [N]} \left(\max_{t \in [T]} \boldsymbol{b}_t(\boldsymbol{x}_t) - \min_{t \in [T]} \boldsymbol{b}_t(\boldsymbol{x}_t)\right)$$
$$= \left(1 - A_{min}^{T-1}\right) \max_{n \in [N]} \max_{t,s \in [T]} (\boldsymbol{b}_t(\boldsymbol{x}_t) - \boldsymbol{b}_s(\boldsymbol{x}_s))$$
$$\leq \left(1 - A_{min}^{T-1}\right) \max_{t,s \in [T]} \|\boldsymbol{b}_t(\boldsymbol{x}_t) - \boldsymbol{b}_s(\boldsymbol{x}_s)\|_\infty,$$

where we exchange the two maximum in the last step. □

Finally, we prove an auxiliary result we used before. This result generalizes Lemma 3 in Wu et al. (2024a) in the scenario when $\alpha_t + \Delta_t < 1$ and $m \leq 0 \leq M$.

*Proof of Lemma D.6.* We first prove the side that is "less than". Note that the desired inequality trivially holds for the base case $s_1$. Then we conduct the following inductive steps. Suppose the desired inequality holds for some $r \in [T - 1]$, namely:

$$s_r \leq A_{min}^{r-1}pm + (1 - A_{min}^{r-1}p)M.$$

Then we show that for time step $r + 1 \in [T]$,

$$s_{r+1} = A_{r+1}s_r + \Delta_{r+1}z_{r+1}$$
$$\leq A_{r+1}\left(A_{min}^{r-1}pm + (1 - A_{min}^{r-1}p)M\right) + \Delta_{r+1}M \quad (15)$$
$$\leq A_{r+1}\left(A_{min}^{r-1}pm + (1 - A_{min}^{r-1}p)M\right) + (1 - A_{r+1})M \quad (16)$$
$$\leq A_{min}\left(A_{min}^{r-1}pm + (1 - A_{min}^{r-1}p)M\right) + (1 - A_{min})M \quad (17)$$
$$= A_{min}^r pm + (1 - A_{min}^r p)M,$$

where we substitute the upper bounds of $s_t$ (by the inductive hypothesis) and $z_t$ in Eq. 15, Eq. 16 holds because either $\Delta_{r+1} = 1 - A_{r+1}$ (by Assumption (i)) or $\Delta_{r+1} \leq 1 - A_{r+1}$ and $M \geq 0$ (by Assumption (ii)), Eq. 17 is satisfied as we are lowering down the coefficient for the smaller end in the convex combination. This concludes the proof by induction.

The proof for the "greater than" side follows from a symmetric argument. We provide a brief proof for completeness. The base case $s_1$ trivially satisfies the desired inequality. Consider an inductive step where $s_r$ for some $r \in [T - 1]$ also satisfies the desired inequality. Then we have:

$$s_{r+1} = A_{r+1}s_r + \Delta_{r+1}z_{r+1}$$
$$\geq A_{r+1}\left((1 - A_{min}^{r-1})qm + A_{min}^{r-1}qM)\right) + \Delta_{r+1}m$$
$$\geq A_{r+1}\left((1 - A_{min}^{r-1})qm + A_{min}^{r-1}qM)\right) + (1 - A_{r+1})m$$
$$\geq A_{min}\left((1 - A_{min}^{r-1})qm + A_{min}^{r-1}qM)\right) + (1 - A_{min})m$$
$$= (1 - A_{min}^r q)m + A_{min}^r qM,$$

which concludes the proof by induction. □

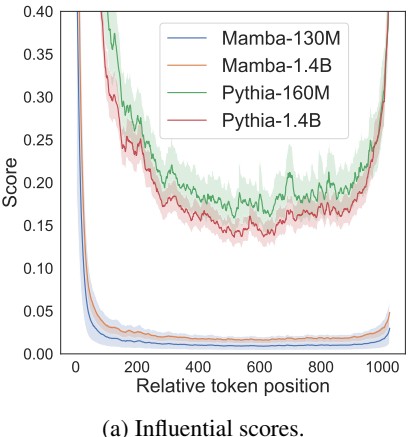
(a) Influential scores.

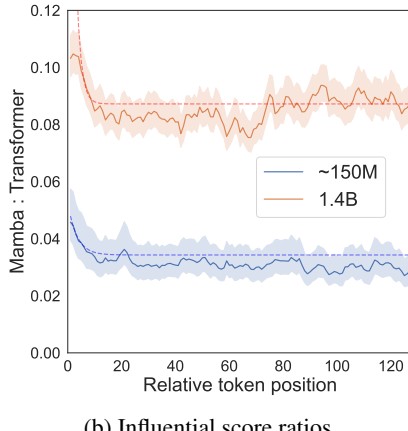
(b) Influential score ratios.

Figure 7: Visualization of influential scores for Mamba and Transformer models under ∼150M and ∼1.4B parameters. (a) illustrates the influence score on the last output token by the preceding input tokens. (b) shows Mamba-to-transformer ratios of influential scores within the nearest 128 tokens.

## E    EXTENDED EXPERIMENTS

### E.1    INFLUENTIAL SCORES: TRANSFORMERS VS. MAMBA

We further compare how influence scores vary with the relative distance between Transformers and Mamba. For our study, we select Pythia (Biderman et al., 2023), which integrates Rotary Position Encoding (RoPE) (Su et al., 2024) into the attention computation. Fig. 7a illustrates the influence scores of each token contributing to the last token. Pythia demonstrates the well-known "lost in the middle" pattern, where tokens at the beginning and end have high influence scores, while those in the middle exhibit lower scores. In contrast, Mamba's highly influential tokens are more concentrated in the nearest positions.

We also compare the decay rate of influence scores for these two models at local tokens. Each curve as shown in Figure 7b represents the ratio of Mamba's influence scores to those of Pythia of various model sizes. The dotted curves are fitted with an exponential function. Our findings reveal that Mamba exhibits an exponentially faster decay speed compared to Pythia. This suggests that Mamba places greater emphasis on local tokens than transformers.

### E.2    NEEDLE IN A HAYSTACK TEST

Our experiments of testing positional bias for Mamba in Sec. 3.2 are based on an open-source project `LLMTest_NeedleInAHaystack` [3]. To validate the retrieval capability of the models while preventing them from relying on memorized information stored in their model weights, we carefully design the inserted statements to contain factual errors. Several examples of such statements are provided in Figure 8. For instance, we insert the statement, "The capital of France is Madrid." and then test the model's retrieval ability by asking the question, "What is the capital of France?" While the correct answer, Paris, is likely memorized by the LLM, if the model "correctly" outputs Madrid based on the context provided, it demonstrates that the model is successfully using the contextual information rather than relying on pre-existing knowledge. This approach ensures that the evaluation focuses on the model's ability to retrieve and process information from the input context. We also add an instruction "ignore the fact, only use information in the context to answer the questions" to facilitate this behavior. We provide the fine-grained visualization result in Figure 9.

### E.3    CIFAR-10 IMAGE CLASSIFICATION

Here we present experiment details in Sec. 3.3, where we conduct image classification on the CIFAR-10 dataset to study locality bias in SSMs. Specifically, $32 \times 32$ RGB images in the dataset are

---

[3] https://github.com/gkamradt/LLMTest_NeedleInAHaystack

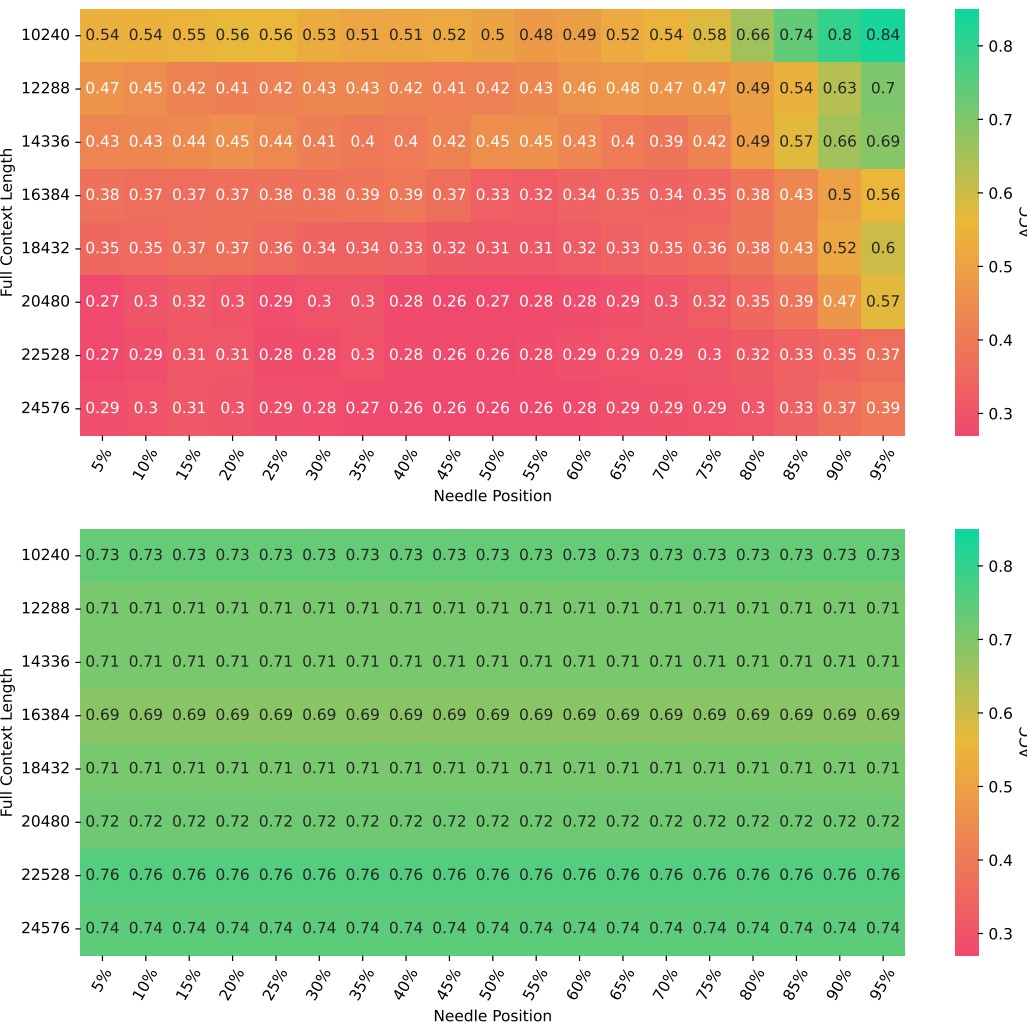

Figure 8: An illustration of our synthetic data.

Figure 9: Detailed comparison between SSM and Transformer on the "Needle in a Haystack" benchmark. The upper figure shows the retrieval accuracy of the Mamba-Codestral-7B model, while the lower figure presents the retrieval accuracy of the Mistral-7B model. We present a heatmap where "full context length" refers to the total length of the document, and "needle position" denotes the relative position of the statement to be retrieved within the context.

flattened into sequences with a shape of $(1024, 3)$, where 1024 represents the sequence length and 3 corresponds to the RGB channels of the pixel tokens. These pixel tokens are then linearly projected into $H$-dimensional features, which are then input into SSM or transformer mixers. In addition to pixel tokens, we insert a class token at the last position of the input sequence. The output state of the class token will be processed by a one-layer classifier head to generate the final logits.

Note that while the ViT architecture (Dosovitskiy et al., 2020) places the class token at the first position of the input sequence, this design is incompatible with SSMs, which rely on causal sequence modeling. In SSMs, the class token must be positioned last to aggregate features from the entire sequence. We position the class token as the last token to establish long-range dependencies between

Table 3: Extended results of adversarial attack experiments on the CIFAR-10 dataset. Classification accuracy is used as the metric.

| Models | (no corrupt) | **Corrupted region** (seq. length = 1024) | | | | | |
| | | [1014:1024] | [0:10] | [768:1024] | [0:256] | [512:544] | [480:576] |
|---|---|---|---|---|---|---|---|
| H3 | 0.654 | 0.629 | 0.654 | 0.394 | 0.639 | 0.603 | 0.543 |
| Transformer | 0.580 | 0.571 | 0.500 | 0.249 | 0.263 | 0.498 | 0.347 |
| RWKV | 0.474 | **0.194** | 0.470 | **0.107** | 0.448 | 0.405 | 0.392 |
| Mamba | 0.674 | 0.348 | 0.664 | **0.099** | 0.597 | 0.515 | 0.446 |

global image features and the leading pixel tokens. Alternative methods for aggregating features across the entire sequence, such as mean pooling (Gu et al., 2021a; Tay et al., 2020) or placing the class token in the middle of the sequence (Zhu et al., 2024), work more robustly in general but do not fit the needs for our arguments on locality.

In addition, our image classification setup differs from Tay et al. (2020), where an 8-bit pixel intensity lookup table is used as the token embedder. Instead, we employ a linear projection to map RGB pixel colors into $H$-dimensional features.

For a fair comparison, the same hyperparameters are used across all models: learning rate = 0.001, weight decay = 0.1, number of layers = 3, feature dimension $H = 32$, and number of states = 64. Each model is trained for 100 epochs. The models and training pipelines are built on Arora et al. (2023). No perturbations are imposed on the input sequences in the training stage.

**Adversarial Attack.** To introduce perturbations to test data for adversarial attack, we first define a corruption length $K$, which is small relative to the entire sequence length. We then replace the leading and trailing $K$ tokens with random Gaussian noise. In our experiments, $K$ is set to 32 and 96, corresponding to one row and three rows of pixels, respectively. Table 3 shows more results under other corruption regions.

**Target Attack.** For the target attack experiments, a target class is first selected. For each image from the other classes, an image from the target class is randomly sampled, and its leading and trailing pixels are used to replace the corresponding pixels in the original image. We test two attack ratios: 256/1024 and 480/1024. Replacing fewer than 256 pixels generally does not result in considerable success rates based on our trials. In our main text, we show success rates when "horse" is the target class. Similar patterns are also observed across other classes. Fig. 10 shows the average success rates obtained by setting each class as the target.

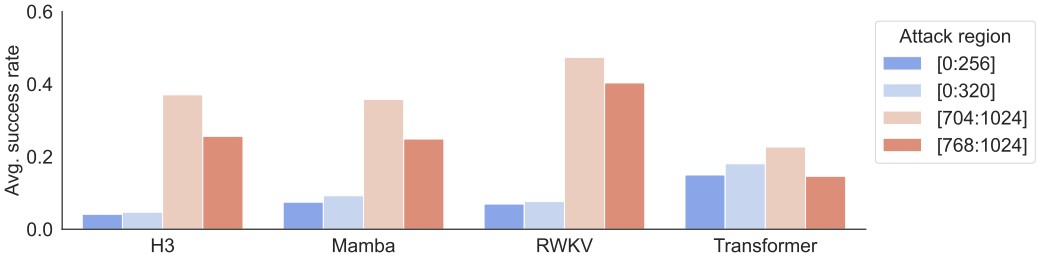

Figure 10: Overall success rate of our target attack experiments on CIFAR-10, calculated by averaging the attack success rates obtained when each class is individually set as the target class.

### E.4 SCALING LAW WITH VARYING DEPTH

In this section, we elaborate on experiment details for Sec. 4.1. We perform casual language modeling (Radford et al., 2018) with Mamba and report the validation perplexity. For each curve in Fig. 4, we fix the depth of the model and vary the number of parameters from 140M to 550M by adjusting the hidden dimension accordingly. The depth is chosen from {16, 24, 32, 48, 64, 72}. All other model configurations keep the default values: the memory state size $N = 16$, the intermediate SSM

dimension is two times the hidden dimension, and the intermediate rank for predicting time step $\Delta_t$ is the hidden dimension divided by 16. We adopt `DataComp-LM` (Li et al., 2024) as our training corpus for its high data quality. The evaluation set is created by holding out a subset of 10M tokens from the training data. We follow the training recipe provided in Appendix E.2.1 of Gu & Dao (2023), which roughly follows the Chinchilla scaling laws (Hoffmann et al., 2022) to allocate the number of tokens for each model size. We test two block sizes {2048, 8192}. During training, we fix the number of tokens in one training batch (# sequences $\times$ sequence length) as 0.5M. The number of training steps is computed by the total number of tokens divided by batch size (# tokens / # tokens in one batch). We use AdamW as the optimizer with gradient clip value 1.0, weight decay 0.1, and linear learning warmup with cosine decay to 1e-5, following Gu & Dao (2023). The peak learning rate is $5\times$ the GPT3 values (Brown et al., 2020) for different model sizes. We summarize the training hyperparameters for varying-sized models in Tab. 4.

| # Params | Training steps | Peak LR | Batch Size (in tokens) | # Tokens |
| --- | --- | --- | --- | --- |
| 100-250M | 4800 | 3e-3 | 0.5M | 2.5B |
| 250-400M | 13500 | 1.5e-3 | 0.5M | 7B |
| 400-550M | 20000 | 1.25e-3 | 0.5M | 10B |

Table 4: Summary of training settings for varying-sized Mamba. The settings are following Chinchilla law (Hoffmann et al., 2022) and consistent with Gu & Dao (2023).

### E.5 SSM POLARIZATION AND ASSOCIATIVE RECALL EXPERIMENTS

In this section, we provide implementation details for our proposed *polarization* technique. Also, we introduce experiment details for the associative recall tasks we used to validate our techniques.

#### E.5.1 POLARIZED MAMBA

Recall that we polarize a channel in $\boldsymbol{A}_t$ with constant zero and another with constant one. Note that in the implementation of Mamba, $\boldsymbol{A}_t$ is parameterized by $\boldsymbol{A}_t = \exp(\Delta_t \boldsymbol{A})$, where $\boldsymbol{A}$ is a learnable parameters. We change the forward pass of Mamba by prepending a zero and appending a number negatively large enough (i.e., -1000). So the pre- and post-exponential $\boldsymbol{A}$ and $\boldsymbol{A}_t$ become:

$$
\boldsymbol{A} \leftarrow \begin{bmatrix} 0 & & \\ & \boldsymbol{A} & \\ & & -1000 \end{bmatrix}, \quad \boldsymbol{A}_t \approx \begin{bmatrix} 1 & & \\ & \exp(\Delta_t \boldsymbol{A}) & \\ & & 0 \end{bmatrix}
$$

We also study two variants: 0-polarized and 1-polarized Mamba by only pretending or appending one or zero to $\boldsymbol{A}_t$ to single out the effects of one and zero gating, respectively.

The gradient flow also complies with the original one, meaning polarization does not change the optimization dynamics of Mamba. Below we show that the polarized $\boldsymbol{A}_t$ does not influence the gradient in terms of the time interval $\Delta_t$. First of all, let $\boldsymbol{G}_t = \frac{\partial \ell}{\partial \Delta_t \boldsymbol{b}_t(\boldsymbol{x}_t)}^{\top} \frac{\partial \Delta_t \boldsymbol{b}_t(\boldsymbol{x}_t)}{\partial \Delta_t}$, then

$$
\frac{\partial \ell}{\partial \Delta_t} = \boldsymbol{G}_t + \sum_{n=2}^{N-1} \frac{\partial \ell}{\partial (\boldsymbol{A}_t)_{n,n}} \frac{\partial (\boldsymbol{A}_t)_{n,n}}{\partial \Delta_t} + \frac{\partial \ell}{\partial (\boldsymbol{A}_t)_{1,1}} \frac{\partial (\boldsymbol{A}_t)_{1,1}}{\partial \Delta_t} + \frac{\partial \ell}{\partial (\boldsymbol{A}_t)_{N,N}} \frac{\partial (\boldsymbol{A}_t)_{N,N}}{\partial \Delta_t} \frac{\partial \ell}{\partial \Delta_t}
$$

$$
= \boldsymbol{G}_t + \sum_{n=2}^{N-1} \frac{\partial \ell}{\partial (\boldsymbol{A}_t)_{n,n}} \frac{\partial (\boldsymbol{A}_t)_{n,n}}{\partial \Delta_t} + \frac{\partial \ell}{\partial (\boldsymbol{A}_t)_{1,1}} \exp(\Delta_t (\boldsymbol{A}_t)_{1,1}) \underbrace{(\boldsymbol{A}_t)_{1,1}}_{=0}
$$

$$
+ \frac{\partial \ell}{\partial (\boldsymbol{A}_t)_{N,N}} \underbrace{\exp(\Delta_t (\boldsymbol{A}_t)_{N,N})}_{\approx 0} (\boldsymbol{A}_t)_{N,N}
$$

$$
\approx \boldsymbol{G}_t + \sum_{n=2}^{N-1} \frac{\partial \ell}{\partial (\boldsymbol{A}_t)_{n,n}} \frac{\partial (\boldsymbol{A}_t)_{n,n}}{\partial \Delta_t},
$$

where the term $\exp(\Delta_t (\boldsymbol{A}_t)_{N,N})(\boldsymbol{A}_t)_{N,N} \to 0$ is achieved by taking $(\boldsymbol{A}_t)_{N,N} \to -\infty$. The component directly contributed by $\boldsymbol{A}_t$ is not affected regardless of the polarization.

| Configurations | # Layers | Recency | Over-smoothing | # KV Pairs | | | Avg. |
|---|---|---|---|---|---|---|---|
| | | | | 64 | 128 | 256 | |
| Default $\boldsymbol{A}_t$ | 2 | ● | ◕ | 98.38 | 81.81 | 36.00 | 72.06 |
| Default $\boldsymbol{A}_t$ | 4 | ◕ | ● | 99.23 | 82.08 | 33.52 | 71.61 |
| $(\boldsymbol{A}_t)_{1,1} = 1$ | 2 | ○ | ◕ | 99.81 | 94.70 | 56.39 | 83.63 |
| $(\boldsymbol{A}_t)_{N,N} = 0$ | 2 | ● | ○ | 98.41 | 81.35 | 36.55 | 72.10 |
| $(\boldsymbol{A}_t)_{N,N} = 0$ | 4 | ◕ | ○ | 99.74 | 92.20 | 52.21 | 81.38 |
| $(\boldsymbol{A}_t)_{1,1} = 1, (\boldsymbol{A}_t)_{N,N} = 0$ | 2 | ○ | ○ | 99.23 | 95.54 | 54.74 | 83.17 |
| $(\boldsymbol{A}_t)_{1,1} = 1, (\boldsymbol{A}_t)_{N,N} = 0$ | 4 | ○ | ○ | 99.94 | 98.80 | 81.56 | 93.43 |

Table 5: Extension to Tab. 5. We note the extent of locality and over-smoothing for each configuration. We consider 1-polarization mitigates locality most significantly, while deepening architecture only relieves recency mildly but deteriorates over-smoothing. 0-polarization alleviates over-smoothening and unleash the benefits by depth scaling.

### E.5.2 ASSOCIATIVE RECALL EXPERIMENTS

Associative Recall (AR) is the task of retrieving a specific value tied to a given key. This ability is particularly important for in-context learning, where a model leverages previously presented information to adapt to new inputs (Olsson et al., 2022; Arora et al., 2023).

In our experiments, we use Mamba as the representative SSM, following the setup described in Arora et al. (2023). The input sequences to the model consist of two parts. The first part is a set of key-value pairs, randomly sampled from a vocabulary, with keys and values drawn independently. The second part contains random keys, selected from those present in the first part (without repetition), and placed according to a power-law distribution at random. Any remaining slots in the sequence are filled with a zero token. The target sequence contains the values corresponding to the positions of the respective key occurrences, while all other tokens are excluded from loss supervision and accuracy evaluation.

We employ a training strategy with mixed sequence lengths and varying numbers of key-value pairs. Specifically, the total sequence length varies across {64, 128, 256, 512, 1024}, with {12.5%, 25%, 50%} of the sequence allocated to key-value pairs and the remainder to queried keys. Each configuration includes 20,000 examples, resulting in a total of 300,000 training examples. For evaluation, the sequence length is fixed at 1024, while the number of key-value pairs varies among 64, 128, 256. We use a batch size of 128 and a learning rate of 1e-3 throughout the experiments. We provide an extended table for the results in Tab. 5.

