# OpenReview forum: "Understanding and Mitigating Bottlenecks of State Space Models through the Lens of Recency and Over-smoothing"
_ICLR.cc/2025/Conference — ICLR 2025 Poster_

### Official Review · Reviewer_veE3 · 2024-11-01

**Soundness:** 3
**Presentation:** 3
**Contribution:** 3
**Rating:** 6
**Confidence:** 3

**Summary:**

This paper investigates why the neural networks based on state space models (SSMs) are inferior to Transformers for sequence data with long-range dependencies. First, the authors theoretically and experimentally demonstrate that the SSM output’s dependency on input diminishes exponentially with the distance from the output position. They call this phenomenon recency bias. They also reveal through experiments that SSMs are vulnerable to attacks targeting the trailing part of the sequence due to recency bias. Furthermore, they explore the effects of increasing the layers of SSMs and find that while this reduces recency bias, it causes over-smoothing.

**Strengths:**

1. The limitations of SSMs’ capabilities are theoretically examined from multiple perspectives, including recency bias, robustness to attack, and over-smoothing.
2. Since multiple experiments closely linked to the theory are conducted, the paper’s claims are easy to understand.

**Weaknesses:**

1. One of the contributions of the paper is that the authors demonstrate the exponential decay of the input-dependency on the SSMs’ output. However, similar findings have already been established in studies such as [Wang & Xue, 2023] (Theorem 3.13).

    [Wang & Xue, 2023] State-space models with layer-wise nonlinearity are universal approximators with exponential decaying memory. In NeurIPS 2023.

2. The experiments on recency bias, such as those in Figure 2, are limited to a few pre-trained models. As with the experiments in Figure 4, it would be preferable to observe the change in performance across the number of parameter counts. In particular, it should be checked whether recency bias can be reduced when the dimensions of the hidden state and the number of heads are moderately increased.
3. Line 463 suggests a trade-off between A_max (related to recency bias) and A_min (related to over-smoothing). However, it is not trivial whether this is actually a challenge when training SSMs since $A_{\max}\approx 1$ and $A_{\min}\approx 0$ can hold at the same time. To clarify this point, additional experiments would be beneficial, such as observing the dynamics of A_min and A_max during training.

**Questions:**

1. In the experiment in Figure 1, the context length is around 1000, whereas in Figure 2, it exceeds 10,000. In the latter case, would there be any difference between SSMs and Transformers with a context length around 1000?
2. Are there any experimental/theoretical results showing that over-smoothing does not occur in Transformers? Figures 4 and 5 contain experimental results demonstrating over-smoothing in Mamba; is it possible to conduct similar verification for Transformers?

---

> ### Author Response · Authors · 2024-11-22
>
> We sincerely thank Reviewer veE3 for reviewing our work and providing insightful suggestions on connecting our findings with the existing literature and conducting additional experiments to further strengthen our claims. We address your concerns as follows.
>
> **1. Comparison with results in [Wang \& Xue, 2023].**
>
> We thank Reviewer veE3 for pointing out the recent work by [Wang \& Xue, 2023]. Distinct from their findings, our primary contribution lies in analyzing **nonlinearity** and **input-dependent** mechanisms, which serves as the key to the success of modern SSMs [1,2,3]. The approach in [Wang \& Xue, 2023] does not naturally extend to this case, as they assume the linearity of the sequence mixer (cf. Eq. 9 in [Wang \& Xue, 2023]). To the best of our knowledge, our work presents the first counterargument showing that even with input-dependent SSMs, effective context filtering may not be achieved. Instead, these mechanisms impose a strict recency bias, inheriting the limitations of **linear** SSMs as previously highlighted by [Wang \& Xue, 2023]. Our analysis considers both $A_t$ and $b_t$ as input-dependent, aligning with settings in Mamba and other follow-up works [2,3]. Furthermore, we conduct a finer-grained analysis, quantitatively relating the decay rate to the specific values within the input-dependent gating matrix $A_t$. We have summarized this discussion into our related work (Sec. 6) and include a full version in Appendix B.
>
> [1] Gu & Dao, Mamba: Linear-time sequence modeling with selective state spaces
>
> [2] Yang et al. Gated linear attention transformers with hardware-efficient training
>
>
> [3] Arora et al. Zoology: Measuring and improving recall in efficient language models
>
> **2. Experiments in Fig. 2 are limited to a few pre-trained models. Whether increasing hidden dimension and the number of heads can mitigate recency?**
>
> We thank Reviewer veE3 for the suggestion to reproduce our empirical observations across a variety of models. However, we would like to point out that the model used in Fig. 2 is `Mamba-Codestral-7B`, which is currently the state-of-the-art SSM model known to us. At present, we are unable to find an open-source pretrained Mamba model that is larger and more advanced. Nevertheless, we still observe that such industry-level model aligns well with our theoretical findings. In contrast, transformer-based model `Mistral-7B` developed by the same team does not exhibit similar limitations. If the reviewer has any recommendations for models that are larger and more capable of long-context tasks than this baseline, we would greatly appreciate it if you could share them with us.
>
> **3. Training dynamics of $A_{max}$ and $A_{min}$.**
>
> We agree with Reviewer veE3 that $A_{max} \approx 1$ and $A_{min} \approx 0$ can happen at the same time. However, our empirical findings show that these values are largely concentrated within a narrow range. To illustrate this, we visualize the distribution of $(A_{max} - A_{min})$ across different channels in Fig. 6 of our revised manuscript. Each bin represents the proportion of channels whose memory state satisfies $(A_{max} - A_{min})$ being smaller than the corresponding threshold on the x-axis. Notably, over 60% of channels exhibit $(A_{max} - A_{min})$ values smaller than 0.5, indicating that most channels cannot simultaneously achieve $A_{max} \approx 1$ and $A_{max} \approx 0$. Memory representations will inevitably undergo either exponential diminishing or over-smoothing.

---

> ### Author Response · Authors · 2024-11-22
>
> **4. Difference between Mamba and Transformer when the context length is around 1000.**
>
> We use different pre-trained models in Fig. 1 and Fig. 2. For plotting the influential score, we adopt author released `Mamba-130M/1.4B`, while the `Mamba-Codestral-7B` pre-trained by MistralAI is used for the long-context retrieval test. Consequently, these two figures are not directly comparable to each.
> Our observations show that `Mamba-Codestral-7B` performs similarly to transformers when the context length is around 1000, likely because it is well-trained for this range of context lengths. However, assuming that the transformer baseline `Mistral-7B` is pre-trained under similar conditions (a reasonable assumption since both models are trained by the same team), `Mistral-7B` demonstrates a clear advantage over `Mamba-Codestral-7B` when the context length is extended.
>
> We conducted a more controlled experiment by training both Mamba and a transformer on the associative recall task, following Zoology [1]. We selected models of size closer to those in Fig. 1 (few millions) and fixed the context length to 1024. The results, presented in the table below, show that Mamba continues to exhibit significant context loss and lower retrieval accuracy compared to transformers, particularly for tokens at longer distances.
>
>
> | Models | [0:512] | [512:1024] |
> | -------- | -------- | -------- |
> | Transformers | 98.96     | 99.21     |
> | Mamba     | 60.75     | 95.07     |
>
>
>
> [1] Arora et al., Zoology: Measuring and improving recall in efficient language models
>
> **5. Does over-smoothing occur to transformers.**
>
> As highlighted in numerous prior works, over-smoothing is also a known issue in transformers [1,2,3,4]. While the theory of over-smoothing in transformers has been extensively studied, our work is the first to theoretically establish the over-smoothing effect in SSMs. Compared to the smoothening rate of causal attention in [4], we show that the feature sharpness (measured by Dirichlet energy, as defined in Sec. 4.2) in SSMs decays at a rate of $O((1-A_{min}^T)^L)$, which is faster than transformers' rate of $O((1-A_{min}^T)^{L/T})$, where we denote $L$ as the number of layers. Empirically, we validate this theoretical result by visualizing token smoothness in our revised Fig. 5, which shows that the pairwise token differences in SSMs decays more rapidly, highlighting a more pronounced over-smoothing issue in SSMs compared to transformers.
>
> Moreover, due to the inherent locality of SSMs (Theorem 3.1 in revision), achieving effective long-range interactions necessitates deeper architectures. In contrast, transformers, which allow arbitrary long-range interactions among tokens, do not have the same requirement. Consequently, depth-scaling limitations are more critical for SSMs than for transformers. We have included an extended discussion into our revised Appendix B.
>
>
> [1] Dong et al., Attention is not all you need: Pure attention loses rank doubly exponentially with depth
>
> [2] Shi et al., Revisiting over-smoothing in bert from the perspective of graph
>
> [3] Wang et al., Anti-oversmoothing in deep vision transformers via the fourier domain analysis: From theory to practice
>
> [4] Wu et al., On the Role of Attention Masks and LayerNorm in Transformers

---

> > ### Author Response · Authors · 2024-11-30
> >
> > Dear Reviewer veE3:
> >
> > We would like to sincerely thank you again for the meaningful suggestions provided in your reviews. As a follow-up, we kindly remind you that the discussion period is ending soon. We hope to use this open response period to fully address your concerns and further improve the quality of our paper. In our revision, we have thoroughly incorporated your suggestions:
> >
> > 1. To better position our work in the literature, we have added a comprehensive comparison with [Wang & Xue, 2023] in Sec. 7 and Appendix B.
> > 2. Additionally, we have included feature smoothness plots in Fig. 5 and connected our theory to the over-smoothing theory of transformers in Appendix B.
> > 3. We have also empirically verified that $A_{min}$ and $A_{max}$ are often ill-distributed, as shown in Fig. 6. Your meaningful question has also guided us to further propose a unified solution in Sec. 6 that mitigates both locality and over-smoothing issues based on our theory. This proposal has been validated through associative recall experiments, as presented in Tab 2.
> >
> > We sincerely look forward to your further feedback and would be happy to provide additional information or clarification if needed. We hope that our paper, which reveals theoretical properties of modern SSMs with practical implications, can be valued and potentially considered for a more positive assessment.
> >
> > Best,
> >
> > Authors of Paper 1195

---

> ### Comment · Reviewer_veE3 · 2024-12-01
>
> Thank you for your comments and for the reminder. I am also grateful for your additional experimental results. Since many of my concerns were resolved by the authors' reply, I raised my score by one.

---

### Official Review · Reviewer_LroV · 2024-11-02

**Soundness:** 3
**Presentation:** 3
**Contribution:** 2
**Rating:** 6
**Confidence:** 2

**Summary:**

This paper identifies two issues in State Space Models: the recency bias in long-context modeling and the over-smoothing effect that arises as the model scales. The authors provide a theoretical framework to analyze these challenges and substantiate their findings with empirical evidence.

**Strengths:**

The paper is well-written and easy to follow.

Experiments are extensive and effectively support the theoretical results.

**Weaknesses:**

While the authors have comprehensively identified and examined these two issues, they did not provide any discussion on possible methods for mitigation. Including a brief exploration of potential strategies could further enrich the paper's contributions.

The problem of recency in SSMs appears somewhat similar to the vanishing gradient problem in RNNs mentioned in, e.g., [1]. Could the authors provide a discussion comparing these two phenomena and highlight the novelty of this work?

The experimental details for Section 4.1 are currently lacking, which makes it challenging to assess the validity of the results fully. Additionally, if I understand correctly, the over-smoothing problem is characterized by a performance degradation when additional layers are stacked beyond a certain threshold (as observed with GNNs, see [2]). However, this effect does not appear to manifest in the experiments presented in Section 4.1. Could the authors clarify whether this phenomenon is inherently absent in SSMs or if it might be a limitation of the current experimental setup?



Minor comments:
- Typo in Eq. 4: The indices $i$ and $j$ are likely meant to be $s$ and $t$.
- Typos in Eq. 18 and Eq. 19: the inequality signs are likely to be $ \geq $ and $ \leq $, respectively.


Reference

[1] Y. Bengio, P. Simard and P. Frasconi, "Learning long-term dependencies with gradient descent is difficult," in IEEE Transactions on Neural Networks, vol. 5, no. 2, pp. 157-166, March 1994.

[2] L. Zhao, Y. Song, C. Zhang, Z. Liu, Y. Wang, and H. Lin, "PairNorm: Tackling Oversmoothing in GNNs," in Proceedings of the International Conference on Learning Representations (ICLR), 2020.

**Questions:**

Please see above.

---

> ### Author Response · Authors · 2024-11-22
>
> We are grateful for Reviewer LroV's efforts and time on reviewing our work. We have taken your valuable suggestions and carefully revise our manuscript to incoporate some initial solutions for mitigating the stated problems. Please see our responses to your questions:
>
> **1. Discussion on possible methods for mitigation.**
>
> We thank Reviewer LroV for this thoughtful suggestion. We added the following dicussion on the potential solutions based on our theoretical results.
>
> As our theory suggests, the maximal and minimal values in $A_t$ directly governs the locality and over-smoothing issues in SSMs, respectively. Encouraging the diversity of values in $A_t$ is thus crucial. While $A_{max} \approx 1$ and $A_{min} \approx 0$ could could theoretically occur simultaneously, our new experiments in Fig. 6 reveal that this does not happen in practice, as the values in $A_t$ are often concentrated within a narrow range. A simple yet promising solution to address this issue is to fix one component of $A_t$ as a constant 1, another as a constant 0, and others freely learnable. This approach ensures that one dimension in the state memory consistently focuses on the current token, counteracting over-smoothing by preventing mixing with previous tokens. Simultaneously, another dimension exclusively retains information from past tokens, avoiding locality issues by preserving the complete history. We are currently in the process of empirically validating this simple solution and will notify the reviewer and update the draft as soon as we have the results.
>
> **2. Discussion comparing recency and vanishing gradients.**
>
> We appreciate Reviewer LroV's comments on connecting our theory with vanishing gradients. Vanishing gradients refer to a challenge in RNNs, where backpropagation-based learning is impeded due to gradient magnitudes decaying exponentially over time. The diminishing dependencies among distant tokens are a fundamental cause of this issue [1]. SSMs were initially proposed to address this limitation by explicitly modeling long-range dependencies, as highlighted in [2, 3]. Subsequent work, such as Mamba, extends this approach by adopting their initialization alongside newly proposed selection mechanisms, which are widely believed to enhance these capabilities. **Our Theorem 3.1 (formerly Theorem 3.2) lies in theoretically challenging this claim.** We demonstrate that modern SSMs still suffer from the recency bias, which not only undermines their ability to capture long-term dependencies but also potentially exacerbates the vanishing gradient problem. We have included this discussion to Appendix B of our revision.
>
> [1] Bengio et al., Learning long-term dependencies with gradient descent is difficult
>
> [2] Gu et al., Hippo: Recurrent memory with optimal polynomial projections
>
> [3] Gu et al., Combining recurrent, convolutional, and continuous-time models with linear state space layers.
>
> **3. The experimental details for Section 4.1 are lacking.**
>
> Our scaling experiments are aligned with the setting adopted in Mamba (see Appendix E.2.1 in [1]). Specifically, in each curve of Fig. 4, we vary the size of models from 140M to 550M by changing the hidden dimension. Then we train these varying-sized Mamba with a certain amount of tokens determined by Chinchilla laws. We have included a detailed version in our Appendix D.3 of the revised draft.
>
> [1] Gu & Dao, Mamba: Linear-time sequence modeling with selective state spaces
>
>
> **4. Performance downgradation caused by oversmoothing does not appear in Section 4.1.**
>
> Over-smoothing does not always result in performance degradation. In many prior studies [1,2,3], over-smoothing has been observed to manifest as performance saturation, where the model's performance plateaus as its depth increases.
>
> In our new experiments, we further increased the depth of Mamba, following the setup described in Sec. 4.1. We observed a noticeable performance drop when the depth exceeded 48 layers. As shown in Fig. 4 (of the revision), the validation perplexity begins to rise for both models with sequence lengths of 2048 and 8192, indicating a decline in performance (e.g., in the 64-layer and 72-layer models).
>
>
> [1] Zhou et al., Deepvit: Towards deeper vision transformer
>
> [2] Wang et al., Anti-oversmoothing in deep vision transformers via the fourier domain analysis: From theory to practice
>
> [3] Chen et al., Bag of tricks for training deeper graph neural networks: A comprehensive benchmark study
>
> **5. Typos and writing errors.**
>
> We thank Reviewer LroV for pointing out writing errors. We have resolved them in our revision. We have moved Lemma 3.1 (currently Lemma C.1) to Appendix C.1 to save some space for other more meaningful discussions.

---

> > ### Author Response · Authors · 2024-11-26
> > **Empirical Validation of Mitigation Solution**
> >
> > We have completed the empirical evaluation of our proposed approach. To tackle the challenges of locality and over-smoothing, we constrain one component of $A_t$ to a constant value of 1, another to 0, while allowing the rest to remain learnable. We refer to this method as polarization, as it effectively enforces one channel to focus entirely on 1 and another on 0.
> >
> > To demonstrate the effectiveness of the polarization method, we employ the associative recall task. In this setup, Mamba models are trained on sequences of key-value pairs to retrieve the value corresponding to a query from its context, following the configuration in [1]. The ability of an SSM to handle long contexts is measured by its accuracy in recalling information as the number of key-value pairs increases. We evaluate Mamba’s performance across configurations with neither, one, or both zero- and one-polarized channels, as well as varying numbers of layers. The results are detailed in the table below.
> >
> > | Configurations                              | # Layers | Locality | Over-smoothing | 64 |  128 | 256 | Avg.   |
> > |---------------------------------------------|----------|----------|----------|----------------|-----------------|-----------------|--------|
> > | Default $A_t$                         | 2        | ✔ | - |98.38          | 81.81           | 36.00           | 72.06  |
> > | Default $A_t$                         | 4        | ✖ | ✔ | 99.23          | 82.08           | 33.52           | 71.61  |
> > | $(A_t)_{1,1} = 1$                     | 2        | ✖ | - | 99.81          | 94.70           | 56.39           | 83.63  |
> > | $(A_t)_{N,N} = 0$                     | 2        | ✔ | ✖ | 98.41          | 81.35           | 36.55           | 72.10  |
> > | $(A_t)_{N,N} = 0$                     | 4        | ✖ | ✖ | 99.74          | 92.20           | 52.21           | 81.38  |
> > | $(A_t)_{1,1} = 1, (A_t)\_{N,N} = 0$ | 2        | ✖ | - | 99.23          | 95.54           | 54.74           | 83.17  |
> > | $(A_t)_{1,1} = 1, (A_t)\_{N,N} = 0$ | 4        | ✖ | ✖ | 99.94          | 98.80           | 81.56           | 93.43  |
> >
> >
> > The main finding is that the default parameterization of $A_t$ faces significant challenges in retrieving information from long contexts (also justify our Theorem 3.1), with deeper architectures further exacerbating the performance, likely due to over-smoothing (also aligned with Theorem 4.2). Introducing a channel polarized to one, however, enables even a shallow Mamba model to achieve high-accuracy associative recall (row 3). Furthermore, when a zero-polarized channel is introduced, deepening the architecture improves performance by addressing the over-smoothing problem (row 5), aligned with our arguments in Sec. 4.1 and 4.2. Most importantly, the combination of both one- and zero-polarized channels, in conjuction with a deeper architecture, achieves the highest performance among all tested configurations (row 7).
> >
> > We once again express our sincere gratitude to Reviewer LroV for their time and effort in reviewing our work. We hope our new results could assist Reviewer LroV in finalizing the final assessment of our hard work, hopefully more positively.
> >
> > [1] Arora et al., Zoology: Measuring and Improving Recall in Efficient Language Models

---

> > > ### Comment · Reviewer_LroV · 2024-11-27
> > >
> > > I would like to thank the authors for their detailed response.
> > >
> > > As my concerns have been sufficiently addressed, I raise my score to 6.

---

### Official Review · Reviewer_53Ds · 2024-11-03

**Soundness:** 4
**Presentation:** 4
**Contribution:** 2
**Rating:** 6
**Confidence:** 5

**Summary:**

This paper lists inspects and provides a theoretical investigation of some of the pitfalls of SSMs. Such pitfalls are caused by a "recency bias," which alters the "token importance," making it flow exponentially down to zero. This fact presents an issue for robustness, as well as for needle-in-a-haystack problems.

**Strengths:**

What I really like about this paper is that it is easy to understand, it is well written and interesting. The contributions are, in my opinion

1) giving practical examples of failure cases of SSMs.

2) showing high-quality and clear plots that can be used for future research or for reference.

This paper indeed seems a bit like a "first step". I like it, though: definitely SSMs have such problems (as intuitively the case and also as pointed out in recent literature) -- yet is **very good to have a reference** - will be interesting for future research!

**Weaknesses:**

I think the following are weaknesses in the current status:

1) The derivation of "exponential memory" results are not novel nor surprising: it is very well known that SSMs have exponential memory (e.g., https://openreview.net/forum?id=i0OmcF14Kf). It is also known that they might struggle with long memory (https://arxiv.org/abs/2402.01032) and to retrieve past information (https://arxiv.org/abs/2312.04927) -- yet scale helps (https://arxiv.org/pdf/2406.07887).

2) Note that exponential memory per see does not imply loss in perplexity and length generalization; this was shown in recent papers such as https://arxiv.org/pdf/2410.11135v1 (or the xLSTM paper https://arxiv.org/abs/2405.04517). Note also that Alibi positional embedding induce a similar exponential decay.

3) The CIFAR robustness example I find a bit weak because here you are implicitly constructing a 1D model for 2D data: in 1D data, recency bias is actually helpful! In 2D there is no notion of recency. I think the example is interesting for improving or researching Visual state-space models such as VMamba. Please frame this a bit more carefully.

3) The paper does not provide hints into solving the stated problems.

**Questions:**

What do you this should the path ahead be for improving SSM models?

---

> ### Author Response · Authors · 2024-11-22
>
> We appreciate Reviewer 53Ds's for the positive initial assessment for our work. Regarding your concerns on novelty and experiment settings, please see our responses to your questions:
>
> **1. The "exponential memory" results are not novel or surprising.**
>
> While the concept of "exponential memory" may seem intuitive to many researchers in this field, we emphasize that our work is the first to theoretically demonstrate this property in the context of **input-dependent mechanisms** such as selection [1], contextualized gating [2], and input-dependent convolution [3], which are integral to modern SSMs. Existing works [3, 4, 5, 6] provide valuable insights into SSMs from various perspectives. However, the results in [4] are limited to **linear SSMs** (e.g., S4), considering nonlinearity only after applying the linear SSMs. The method proposed in [4] does not naturally extend to this scenario, as it assumes linearity in the sequence mixer (cf. Eq. 9 in [4]). Please refer to Appendix B of our revision for a detailed comparison. [3] and [5] primarily analyze the limitations of SSMs using specific synthetic tasks, such as copying and associative recall, which do not directly expose the long-range dependencies among tokens. [6] establishes a strong benchmark for modern SSMs. Our theoretical results further elucidate these limitations. Additionally, our long-context retrieval experiments extend these findings to scaled-up industrial-level models, demonstrating that even with scaling-up, the long-range limitations of SSMs persist. Furthermore, we identify a novel security vulnerability in SSMs, which can lead to easier jailbreaks in LLMs (see additional discussions in Sec. 3.3 of our revision). To cover these points, we have added a new related work section in our revision.
>
>
> [1] Gu & Dao, Mamba: Linear-time sequence modeling with selective state spaces
>
> [2] Yang et al. Gated linear attention transformers with hardware-efficient training
>
>
> [3] Arora et al. Zoology: Measuring and improving recall in efficient language models
>
> [4] Wang & Xue, State-space models with layer-wise nonlinearity are universal approximators with exponential decaying memory
>
> [5] Jelassi et al., Repeat After Me: Transformers are Better than State Space Models at Copying
>
> [6] Waleffe et al., An Empirical Study of Mamba-based Language Models
>
> **2. Exponential memory does not imply loss in perplexity and length generalization.**
>
> We agree with Reviewer 53Ds that exponential memory does not necessarily lead to a loss in perplexity and can even aid in length generalization, as demonstrated by xLSTM [1] and Alibi [2]. As discussed in Sec. 3.1, recency serves as a near-optimal bias for language modeling, meaning its imposition does not negatively impact perplexity.
>
> However, we wish to emphasize that a key contribution of our work is showing that, despite not affecting perplexity, this decaying design has notable impacts on many downstream tasks. These issues include, but are not limited to, substantial loss of long-distance context and more severe security vulnerabilities (refer to our responses to (3)). This observation also underscores that validation perplexity may not fully reflect all aspects of a model's performance and is often an inadequate metric for evaluating long-range dependencies.
>
> [1] Pöppel et. al., xLSTM: Extended long short-term memory
>
> [2] Press et. al., Train short, test long: Attention with linear biases enables input length extrapolation

---

> ### Author Response · Authors · 2024-11-22
>
> **3. CIFAR robustness examples are weak and needs reframe.**
>
> The image classification with CIFAR is used as a showcase to demonstrate the robustness issues of Mamba caused by recency. We note that testing sequential models with image datasets is a standard practice in sequence modeling, as used in well-known works [1, 2]. Image dataset is also a part of Long-Range Arena (LRA) [3], a popular benchmark for long-sequence modeling.
>
> While our adversarial attack experiments focus on image datasets, the findings extend to language models. In particular, it is reasonable to generalize our conclusions on targeted attack experiments to LLMs that: SSM-based LLMs can be particularly vulnerable to jailbreak attacks [4, 5]. This is because SSMs prioritize recent information over past tokens, making it easier to bypass system prompts by appending jailbreak instructions at the end of the input.  Moreover, our theoretical analysis suggests that fine-tuning LLMs with instructional datasets or human feedback to enforce adherence to system prompts may not resolve this vulnerability, as the recency bias remains inherent to SSM models regardless of weight configurations. We have reframed our Sec. 3.3 to include these discussions.
>
> [1] Gu et al., Combining Recurrent, Convolutional, and Continuous-time Models with Linear State Space Layers
>
> [2] Gu et al., Efficiently modeling long sequences with structured state spaces
>
> [3] Tay et al., Long range arena: A benchmark for efficient transformers
>
> [4] Perez & Ribeiro, Ignore previous prompt: Attack techniques for language models
>
> [5] Zou et al., Universal and transferable adversarial attacks on aligned language models
>
> **4. No hint to solve the state problems.**
>
> We appreciate Reviewer 53Ds for suggesting the inclusion of hints to address the stated problems. While this aspect was initially beyond the scope of our work, our theoretical findings do point to some intriguing potential solutions worth exploring.
>
> Our theory highlights that the maximum and minimum values in $A_t$ play a pivotal role in determining the extent of locality and over-smoothing in SSMs, respectively. Therefore, promoting diversity in the values of $A_t$ is a potential approach of mitigating locality and over-smoothing simultaneously. Although it is theoretically possible for $A_{max} \approx 1$ and $A_{min} \approx 0$ to happen at the same time, our new experiments in Fig. 6 reveal that, in practice, the values in $A_t$ tend to cluster within a narrow range. To mitigate this issue, a straightforward yet effective strategy involves fixing one component of $A_t$ at a constant 1, another at a constant 0, while allowing the remaining components to be freely learnable. This design ensures that one dimension of the state memory (with all-pass gating value) focuses exclusively on retaining past information, addressing locality concerns by preserving the full history. The dimension with zero gating value remains dedicated to the current token, thereby mitigating over-smoothing by preventing unnecessary mixing with previous tokens. We are actively validating this approach empirically and will update the draft with results and inform the reviewer as soon as we have the results.

---

> > ### Author Response · Authors · 2024-11-26
> > **Empirical Validation of Proposed Solution**
> >
> > We have conducted empirical validation of our proposed solution. Specifically, we address locality and over-smoothing issues by fixing one component of $A_t$ to a constant 1, another to a constant 0, while allowing the remaining components to be freely learnable. We term this approach *polarization*, as it polarizes one channel to 1 and another to 0.
> >
> > To validate the effectiveness of the polarization technique, we use the associative recall task. In this task, Mamba models are trained on sequences of key-value pairs to retrieve the value associated with a query based on its context, following the setup in [1]. The long-context capability of an SSM is evaluated by its ability to recall information accurately as the number of key-value pairs increases. We assess Mamba’s performance with configurations that include neither, one of, or both zero- and one-polarized channels, across various numbers of layers. The results are summarized in the table below.
> >
> > | Configurations                              | # Layers | Locality | Over-smoothing | 64 |  128 | 256 | Avg.   |
> > |---------------------------------------------|----------|----------|----------|----------------|-----------------|-----------------|--------|
> > | Default $A_t$                         | 2        | ✔ | - |98.38          | 81.81           | 36.00           | 72.06  |
> > | Default $A_t$                         | 4        | ✖ | ✔ | 99.23          | 82.08           | 33.52           | 71.61  |
> > | $(A_t)_{1,1} = 1$                     | 2        | ✖ | - | 99.81          | 94.70           | 56.39           | 83.63  |
> > | $(A_t)_{N,N} = 0$                     | 2        | ✔ | ✖ | 98.41          | 81.35           | 36.55           | 72.10  |
> > | $(A_t)_{N,N} = 0$                     | 4        | ✖ | ✖ | 99.74          | 92.20           | 52.21           | 81.38  |
> > |  $(A_t)_{1,1} = 1, (A_t)\_{N,N} = 0$ | 2        | ✖ | - | 99.23          | 95.54           | 54.74           | 83.17  |
> > |  $(A_t)_{1,1} = 1, (A_t)\_{N,N} = 0$ | 4        | ✖ | ✖ | 99.94          | 98.80           | 81.56           | 93.43  |
> >
> > The key observation is that the default parameterization of $A_t$ struggles with information retrieval from long contexts, and increasing the depth of the architecture further degrades performance, likely due to over-smoothing. However, introducing a channel polarized to one enables even a shallow Mamba model to achieve high-accuracy associative recall (row 3). Additionally, performance improves by deepening Mamba when the over-smoothing issue is mitigated through the adoption of a zero-polarized channel (row 5). Notably, combining both one- and zero-polarized channels while simultaneously deepening the architecture yields the best performance across all configurations (row 7).
> >
> > We again express our gratitude for Reviewer 53Ds' efforts and time on reviewing our work. We hope our new results could help Reviewer 53Ds finalize the final assessment of our hard work.
> >
> > [1] Arora et al., Zoology: Measuring and Improving Recall in Efficient Language Models

---

> ### Author Response · Authors · 2024-11-30
>
> Dear Reviewer 53Ds:
>
> Thank you again for your valuable feedback and thoughtful suggestions during the review process. We have made the following updates based on your feedback:
>
> 1. Added a dedicated related work section (Sec. 7) to discuss relevant prior work and included detailed comparisons with [Wang & Xue, 2023] in Appendix B.
> 2. Expanded Sec. 3.3 to generalize the implications of our robustness experiments to LLM jailbreaking.
> 3. Proposed a unified technique in Sec. 6 to simultaneously mitigate locality and over-smoothing, supported by validations aligning with our theory (Tab. 2). Also see our previous replies.
>
> As the discussion period is nearing its end, have you gotten a chance to read our responses and finalize your assessment? We sincerely wish our hard work, revealing the recency issue and over-smoothing nature of modern SSMs for the first time, could receive a more positive feedback.
>
> Best,
>
> Authors of Paper 1195

---

### Official Review · Reviewer_2oyp · 2024-11-03

**Soundness:** 2
**Presentation:** 3
**Contribution:** 2
**Rating:** 6
**Confidence:** 2

**Summary:**

The paper presents 2 limitations of current SSM. The first points out to the fact that SSMs suffer from a strong recency bias. This in turn limits the SSMs ability to recall distant information and introduces robustness issues. Secondly, it is revealed that as SSMs increase in depth, they exhibit a tendency toward over-smoothing, resulting in token representations becoming increasingly indistinguishable

**Strengths:**

1. The paper presents important findings, which open up future directions to resolve the issue.
2. The section on adversarial attacks is praiseworthy, as it utilizes one of the findings to describe something very practical.
3. Both findings are substantiated with some theoretical understanding.

**Weaknesses:**

1. While, empirical results are satisfactory for the recency of SSMs, I don't feel these are extensive for the claim of over-smoothing in SSMs.
2. How do initialization scheme of SSMs interfere with these findings. Can there be some discussion around these?

**Questions:**

Please see above

---

> ### Author Response · Authors · 2024-11-22
>
> We appreciate Reviewer 2oyp's for the positive initial assessment of our work. We have included more experiments to strengthen our over-smoothing part. Please see our detailed responses below:
>
> **1. Empirical results for over-smoothing are not extensive.**
>
> To strengthen our experiments on over-smoothing, we have added more results in Fig. 4 and Fig. 5. Specifically, we extended the depth-scaling experiments in Sec. 4 and further increased the depth of Mamba, following the setup in Sec. 4.1. We observed a significant performance drop when the depth exceeded 48 layers. As shown in Fig. 4 (of the revision), validation perplexity rises for both models with sequence lengths of 2048 and 8192, indicating a decline in performance (e.g., in the 64-layer and 72-layer models). This finding suggests that over-smoothing caused by increased depth not only saturates performance gains but also degrades performance, akin to previous observations in GNNs [1].
>
> We have additionally measured the smoothness of trasformers' hidden states in Fig. 5. We demonstrate that SSMs are more prone to over-smoothing.
>
> We would greatly appreciate Reviewer 2oyp's detailed advice on how we might further enhance our experimental section.
>
> [1] Li et al., Deeper Insights into Graph Convolutional Networks for Semi-Supervised Learning
>
>
> **2. How do initialization scheme interfere with these findings?**
>
> We thank Reviewer 2oyp for initiating this insightful discussion. Our theoretical results confirm that initialization plays a crucial role, as it impacts the distribution of values in $A_t$, where the maximum and minimum values determine the extent of locality and over-smoothing, respectively. To address both issues, our theory suggests that $A_t$ should span a wider range of values, which could be achieved through carefully designed initialization schemes. However, we note that the current initialization scheme used in Mamba performs poorly in promoting diversity in $A_t$. As shown in our newly added Fig. 6, the majority of values are closely clustered, leading to either short-range dependencies or over-smoothing of token representations.
>
> Moreover, we point out why the initialization scheme derived from HiPPO theory fails to work effectively for Mamba in Sec. 5 (formerly Sec. 3.1). The primary reason is that Mamba simplifies the parameterization to a diagonal form, which undermines the merits of HiPPO matrices. This limitation has also been noted in previous work (Proposition 3 in [1]).
>
> [1] Gu et al., How to train your hippo: State space models with generalized orthogonal basis projections

---

> > ### Comment · Reviewer_2oyp · 2024-11-25
> >
> > Dear authors,
> >
> > Thank you for the rebuttal. My score remains the same

---

### Meta-Review · Area_Chair_uJwe · 2024-12-20

**Metareview:**

This paper presents a comprehensive analysis of the limitations of State Space Models (SSMs) in capturing long-range dependencies in sequential data. The authors theoretically and empirically demonstrate that SSMs suffer from a recency bias, where the model's output dependency on input diminishes exponentially with distance from the output position. Additionally, they show that increasing the number of layers in SSMs can lead to over-smoothing, which degrades the model's performance.

The reviewers have provided thoughtful feedback, and the authors have addressed their concerns through revisions and additional experiments. The authors have clarified the differences between their work and existing studies and have provided more detailed comparisons with Transformers.

Overall, the paper has been significantly improved through the revision process, and the authors have demonstrated a good understanding of the limitations and potential solutions for SSMs. The paper's contributions, including the theoretical analysis of recency bias and over-smoothing in SSMs, are valuable and relevant to the field.

Hence, after discussion with the authors and among themselves, the reviewers find the paper much improved, with all reviewers leaning towards acceptance. We are therefore happy to accept the paper. We would still like to encourage the authors to address all the reviewers' comments in the camera-ready version.

**Additional Comments On Reviewer Discussion:**

see above

---

### Decision · Program_Chairs · 2025-01-22

Accept (Poster)